# PIXEL-AWARE ACCELERATED REVERSE DIFFUSION MODELING

## ABSTRACT

We propose in this paper an analytically new construct of a diffusion model whose drift and diffusion parameters yield a faster time-decaying Signal to Noise Ratio in the forward process. The proposed methodology significantly accelerates the forward diffusion process, reducing the required diffusion time-steps from around 1000 seen in conventional models to 200-500 without compromising image quality in the reverse-time diffusion. Additionally, in a departure from conventional models which typically use time-consuming multiple runs, we introduce a parallel data-driven model to generate a reverse-time diffusion trajectory in a single run of the model. The construct cleverly carries out the learning of the diffusion coefficients via an estimate of the structure of clean images. The resulting collective block-sequential generative model eliminates the need for MCMC-based sub-sampling correction for safeguarding and improving image quality, which further improve the acceleration of image generation. Collectively, these advancements yield a generative model that is at least 4 times faster than conventional approaches, while maintaining high fidelity and diversity in generated images, hence promising widespread applicability in rapid image synthesis tasks.

## 1 INTRODUCTION

Generative diffusion models (GDMs) have recently emerged as powerful tools for image modeling and numerous other applications (Sohl-Dickstein et al., 2015; Song & Ermon, 2019; Ho et al., 2020; Song et al., 2021b), offering exceptional fidelity and generative diversity (Yang et al., 2023). In contrast to existing generative models, like Generative Adversarial Networks (GANs) (Goodfellow et al., 2014) and Variational Autoencoders (VAEs) (Kingma & Welling, 2013), GDMs are more stable in training and less sensitive to hyper-parameter selection (Kingma & Welling, 2019).

Generative diffusion approaches gradually corrupt image data with increasingly random noise in the forward diffusion steps. The noise removal[1] is progressively learned in the reverse diffusion to recover the desired image data to best match its initial distribution. GDMs have been represented as discrete as well as continuous-time processes. Discrete-time processes were initially used in diffusion probabilistic models (Sohl-Dickstein et al., 2015) and later refined as the denoising diffusion probabilistic models (DDPMs) (Ho et al., 2020), The training task was to learn the reverse posterior distribution by maximizing a variational lower bound on the model log likelihood. Later Score-based Generative Models (SGMs) - represented by Noise-Conditional Score Networks (NCSNs) (Song & Ermon, 2019) and (Song et al., 2021b) - were successfully used to achieve high quality image samples through denoising score matching and annealed Langevin dynamics. The continuous-time GDMs are fundamentally based on continuous stochastic differential equations (SDEs), with a continuous forward diffusion model defined to add random noise to an image according to some selected drift and diffusion parameters, and a reverse

---

[1]This may equivalently be viewed as learning the statistical structure of the data for given additive white noise.

diffusion (Anderson, 1982) is run by a reverse-time SDE governed by a score-based model to be learned by deep neural networks.

While effective, the performance of conventional diffusion models entails a slow convergence, with a quality image generation requiring a large number of time-steps, consequently leading to an increased computational complexity. To this end, much effort (Jolicoeur-Martineau et al., 2021a;b; Salimans & Ho, 2021; Zhang & Chen, 2023; Dockhorn et al., 2022; Lyu et al., 2022; Zheng et al., 2022) has been dedicated to reducing this lengthy process and to improving the quality for prediction-correction methods by Markov chain Monte Carlo (MCMC) subsampling & modified Langevin dynamics (Song et al., 2021b). Current models have, however,only focused on reducing the reverse trajectory by employing sub-sampling or fast ODE solver based strategies.

In this paper, we propose an alternative approach and use insights from statistical mechanics of particles to account for local (pixel-level) SNR in driving the microscopic dynamics of the diffusion process. In so doing, our novel diffusion model leverages the structure of clean image data to learn the drift and diffusion parameters at a microscopic level to accelerate the forward diffusion. Specifically, these parameters increase the rate of degradation according to image pixel-SNR in contrast to the uniform regime of existing models. This is inspired by the well-known water pouring algorithm paradigm (Gallager, 1968) employed in multi-channel communication systems. The water pouring algorithm allocates power to a channel in accordance with the noise-level experienced in that channel, with more degraded channels getting more power. Intuitively, one may interpret the macroscopic forward diffusion as a parallel (bundle) process of microscopic forward diffusion processes occurring on individual pixels in parallel. In our model, the forward diffusion scheduling is dependent, as detailed later, on the initial clean pixel values while each pixel maintains its own diffusion independent from others. We demonstrate that by employing this pixel based scheduling strategy, we can achieve the target goal of reaching isotropic Gaussian distribution on all the pixels much faster than the conventional pixel agnostic diffusion scheduling in the forward diffusion stage.

With such an image-aware forward diffusion in hand, we proceed with an autoencoder to learn the combined diffusion schedule across all the pixels of a noisy image. This learned schedule is subsequently used in a data-driven reverse-time diffusion model to generate the complete trajectory of the reverse-time diffusion. While conventional models generate the reverse trajectory one step at a time requiring multiple model runs, we leverage the structural information learned in the scheduling strategy to generate the whole reverse-time diffusion path in one go without compromising on the over-all generated image quality. As a result of this combined strategy, we are able to accelerate the reverse-time diffusion process by at least 4 times in terms of the total generation time. On account of its relatively simple structure, our pixel-wise diffusion has the potential to be incorporated into existing methods to further speed up sampling. As an example, the latent diffusion model (Rombach et al., 2022b), with some special care given to the image scale, should be able to swap the conventional diffusion with our new model. However, we only focus on our novel diffusion model in this paper.

## 2 BACKGROUND

### 2.1 FORWARD DIFFUSION

Associating to a data sample $\mathbf{x}_0 \in \mathbb{R}^d$ distributed as $\mathbf{x}_0 \sim q(\mathbf{x}_0)$, a common forward diffusion (Sohl-Dickstein et al., 2015) is tantamount to defining a Markov chain of samples $\mathbf{x}_1, ..., \mathbf{x}_T$ such that:

$$q(\mathbf{x}_1, ..., \mathbf{x}_T | \mathbf{x}_0) = \prod_{i=1}^{T} q(\mathbf{x}_i | \mathbf{x}_{i-1}), \ \ q(\mathbf{x}_i | \mathbf{x}_{i-1}) = \mathcal{N}(\sqrt{1 - \beta_i} \mathbf{x}_{i-1}, \beta_i \mathbf{I}), \ i \in \{1, ..., T\}, \tag{1}$$

where $\beta_i \in (0,1)$, $\forall\ i \in \{1,...,T\}$, is an increasing scalar schedule starting from a very small positive value progressing towards 1 in $T$ steps. For a sufficiently large, $T$, a well behaved increasing schedule $\beta_i$ ensures that $\mathbf{x}_T \sim \mathcal{N}(\mathbf{0},\mathbf{I})$ . In relation to Eqn. 1, a forward diffusion of this form may follow as a discretized Markovian process,

$$\mathbf{x}_{i+1} = \sqrt{\alpha_i}\mathbf{x}_i + \sqrt{1-\alpha_i}\epsilon_i,\ \epsilon_i \sim \mathcal{N}(\mathbf{0},\mathbf{I}), \tag{2}$$

where $\alpha_i = 1 - \beta_i$, the first addend controls the drift of $\mathbf{x}_{i+1}$, and the second one is its diffusion term. Another iterative form of Eqn. 2 in terms of the initial sample value $\mathbf{x}_0$, can also be obtained as a reparameterized equation(Kingma & Welling, 2013),

$$\mathbf{x}_{i+1} = \sqrt{\overline{\alpha}_i}\mathbf{x}_0 + \sqrt{1-\overline{\alpha}_i}\tilde{\epsilon}_i,\ \tilde{\epsilon}_i \sim \mathcal{N}(\mathbf{0},\mathbf{I}), \tag{3}$$

where $\overline{\alpha}_i = \prod_{k=1}^{i} \alpha_k$ and $\tilde{\epsilon}_i$ is a linear combination of $\epsilon_k \sim \mathcal{N}(\mathbf{0},\mathbf{I})$, $k \in \{1,...,i\}$, such that $\tilde{\epsilon}_i \sim \mathcal{N}(\mathbf{0},\mathbf{I})$. Note that $\beta_i$ controls the diffusion of the process, and a number of its variations have been used including a linear profile (Ho et al., 2020), a harmonic (Sohl-Dickstein et al., 2015) and a squared cosine dependency (Song et al., 2021a). In all these cases, a simple scalar $\beta_i$ is invariably used for each element of the vector $\mathbf{x}_i = [x_i^1,\cdots,x_i^j,\cdots,x_i^d] \in \mathbb{R}^d$, where $d$ is the number of pixels in an image.

## 2.2 Sampling for reverse-time diffusion

While Eqn. 2 implies $q(\mathbf{x}_i|\mathbf{x}_{i-1})$ is an explicitly known conditional Gaussian distribution for $\mathbf{x}_i$, the posterior distribution $q(\mathbf{x}_{i-1}|\mathbf{x}_i)$ is unknown. The probability of $\mathbf{x}_{i-1}$ conditioned on $\mathbf{x}_i$ and $\mathbf{x}_0$ can, however, be explicitly expressed using Bayes rule (Ho et al., 2020) as

$$q(\mathbf{x}_{i-1}|\mathbf{x}_i,\mathbf{x}_0) = \mathcal{N}(\tilde{\mu}(\mathbf{x}_i,\tilde{\epsilon}_i),\tilde{\beta}_i\mathbf{I}),$$

$$\tilde{\mu}(\mathbf{x}_i,\tilde{\epsilon}_i) = \frac{1}{\sqrt{\alpha_t}}\left(\mathbf{x}_i - \frac{1-\alpha_i}{\sqrt{1-\overline{\alpha}_i}}\tilde{\epsilon}_i\right), \tag{4}$$

$$\tilde{\beta}_i = \frac{1-\overline{\alpha}_{i-1}}{1-\overline{\alpha}_i}\beta_i.$$

DDPM based models (Ho et al., 2020; Sohl-Dickstein et al., 2015; Song et al., 2021a; Peebles & Xie, 2022) approximate $q(\mathbf{x}_{i-1}|\mathbf{x}_i,\mathbf{x}_0)$ using $p_\theta(\mathbf{x}_{i-1}|\mathbf{x}_i) = \mathcal{N}\left(\tilde{\mu}\left(\mathbf{x}_i,\varepsilon_\theta(\mathbf{x}_i,i)\right),\tilde{\beta}_i\right)$, where $\varepsilon_\theta(\mathbf{x}_i,i)$ is a neural network model for removing the noise added in the forward diffusion step. To learn to predict the mean $\tilde{\mu}(.)$, a Variational Lower Bound (VLB) on the negative log likelihood $p_\theta(\mathbf{x}_0)$ is maximized. The VLB is given as:

$$L_{\text{VLB}} = \mathbb{E}_q\Big[D_{\text{KL}}\Big(q(\mathbf{x}_T|\mathbf{x}_0)||p_\theta(\mathbf{x}_T)\Big)$$

$$+ \sum_{i=2}^{T} D_{\text{KL}}\Big(q(\mathbf{x}_{i-1}|\mathbf{x}_i,\mathbf{x}_0)||p_\theta(\mathbf{x}_{i-1}|\mathbf{x}_i)\Big) - \log\Big(p_\theta(\mathbf{x}_0|\mathbf{x}_1)\Big)\Big]. \tag{5}$$

The neural network output $\varepsilon_\theta(\mathbf{x}_i,i))$, together with the parameters $\overline{\alpha}_i$, $\beta_i$, enables the reverse-time diffusion trajectory constructed as,

$$\mathbf{x}_{i-1} = \tilde{\mu}(\mathbf{x}_i,\varepsilon_\theta(\mathbf{x}_i,i)) + \sqrt{\tilde{\beta}_i}\sigma_i,\ \sigma_i \sim \mathcal{N}(\mathbf{0},\mathbf{I}),\ i \in \{T,\cdots,1\}. \tag{6}$$

This perspective has led to various GDMs sampling from the posterior distribution.

Additionally, a continuous-time version of Eqn. 2 was given in (Song et al., 2021b) for $T \to \infty$ yielding a forward-time. Relying on (Anderson, 1982; Hyvärinen, 2005), the corresponding reverse-time SDE can be built via learning the so-called score function $\nabla_{\mathbf{x}_t} \log(q(x_t))$ iteratively through through a neural network.

Two main neural network architectures are used for learning tasks: U-Net based models (Song & Ermon, 2019; Ho et al., 2020; Song et al., 2021b;a) and Transformer-based models (Peebles & Xie, 2022; Rombach et al., 2022a). The U-Net (Ronneberger et al., 2015), popular for semantic segmentation tasks, uses a downsampling encoder and an upsampling decoder, with feature maps from the encoder concatenated with inputs of the decoder at different resolutions. Upsampling is a sparse operation. A good prior from earlier stages aids the decoder to better represent the localized features. Newer models use Transformer architectures, which work on lower-dimensional latent encodings instead of images, to offer higher generation quality despite being more computationally intensive (Peebles & Xie, 2022).

## 3 METHODOLOGY: IMAGE AWARE DIFFUSION

### 3.1 MOTIVATION

In the forward direction of a diffusion process, a clean image with $d$ pixels, represented as $\mathbf{x}_0 = [x_0^1, \ldots, x_0^d]$ is diffused iteratively in $T$ steps. With large enough $T$, we get $\mathbf{x}_T \sim \mathcal{N}(\mathbf{0}, \mathbf{I}_d)$, indicating that the diffusion of the various data pixels leads to approximately 0 SNR. The water pouring algorithm employed in multi-channel communication systems(Gallager, 1968), similarly addresses assignment of signal power distribution across frequency channels with different ambient noise powers to maximize SNR. Faced with our objective of all pixels simultaneously achieving approximately 0 SNR over a certain time interval (the total number of steps, $T$), it makes sense to diffuse higher-valued pixels at a faster rate than lower-valued pixels.

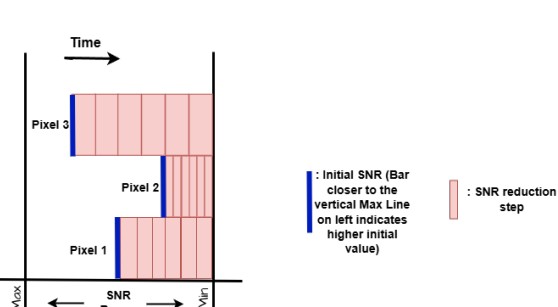

FIGURE 1: A toy example: SNR degradation in forward diffusion direction across three pixels as a dual of water pouring algorithm

Fig. 1 shows as an example SNR reduction across 3 pixels with different SNRs over 6 time-steps. In the beginning, Pixel 3 has the highest value while Pixel 2 has the lowest value. To reach 0 SNR simultaneously in 6 steps, different pixels experience SNR reduction at different rates.

This also implies that our group-diffusion process, will require a vector $\beta_i \in \mathbb{R}^d$, whose elements are different for different pixels $x_i^j$, in $\mathbf{x}_i = [x_i^1, \cdots, x_i^j, \cdots, x_i^d] \in \mathbb{R}^d$ to reflect each diffusion yielding specific drift and diffusion determined by each pixel. To this end, we propose here a carefully chosen image-

aware scheduling forward process which converges to the isotropic standard normal distribution at a faster rate. Consequently the reverse process can converge much faster.

Furthermore, conventional GDMs generate new samples by a reverse diffusion process which involves sequential sampling (over $i$ ranging from $T$ to 1) from the learned conditional posterior distributions, $p_\theta(\mathbf{x}_{i-1}|\mathbf{x}_i)$ with $\mathbf{x}_T \sim \mathcal{N}(\mathbf{0}, \mathbf{I}_d)$. or with a uniformly distributed subsampling sequence, as discussed in Denoising Diffusion Implicit Model (DDIM) (Song et al., 2021a). The sequential sampling entails multiple forward passes through a trained model, significantly increasing the overall generation time. To overcome this difficulty, we proposed here our second innovative contribution - our proposed parallel generation of the reverse diffusion. We rely on a more informed prior, specifically a rough estimate of the clean image $\mathbf{x}_0$. This provides some early-scale feature information of the clean image such as image boundaries. It acts as a regularizer to our model. This, together with our fast pixel-wise diffusion allows us a simultaneous parallel generation of reverse diffusion steps.

### 3.2 Redefining forward diffusion

#### 3.2.1 Definitions

We define image scale $\mathbf{x}_\delta \in \mathbb{R}^d$ as

$$\mathbf{x}_\delta \triangleq e^{-\gamma \mathbf{x}_0}, \tag{7}$$

where $\gamma$ is a scalar hyperparameter such that $x_0^j << \gamma < T, \ \forall \ j \in \{1, \cdots, d\}$ and exponentiation is done element-wise.

The diffusion schedule parameters now become vectors, redefined as

$$\boldsymbol{\alpha}_i = \mathbf{1} - \boldsymbol{\beta}_i = \mathbf{x}_\delta^{1/T}. \tag{8}$$

where $\mathbf{1} = [1, 1, \cdots, 1]$ is a d-dimensional vector. This allows the diffusion schedule to vary across all the pixels $x_i^j, \ j \in \{1, ..., d\}$. This is in contrast to conventional Denoising Diffusion Probabilistic Model (DDPM) (Ho et al., 2020), in which the schedule parameters can be regarded as vectors with the same repeated element (ours being distinct): $\boldsymbol{\alpha}_{C,i} = \mathbf{1} - \boldsymbol{\beta}_{C,i} = \alpha_i \mathbf{1}$, where $\alpha_i$ is a scalar independent of $\mathbf{x}_0$. The resulting reparametrized forward step dependent on $\mathbf{x}_0$ can be written as

$$\mathbf{x}_i = \sqrt{\overline{\boldsymbol{\alpha}_i}} \odot \mathbf{x}_0 + \sqrt{1 - \overline{\boldsymbol{\alpha}_i}} \odot \tilde{\epsilon}_i, \ \overline{\boldsymbol{\alpha}}_i = e^{-\gamma \ i \mathbf{x}_0/T}, \ i \in \{0, 1, ..., T\} \tag{9}$$

where $\odot$ is an element-wise multiplication.

We further assume that all $x_0^j$ are normalized to the range $(0, 1]$, with small scalar value added to all $x_0^j$, so that none of the resulting pixels is exactly 0 to ensure that $e^{-\gamma \ i \ \mathbf{x}_0/T}$ vary with $i$. From Eqn. 9, it is clear that as $i$ increases from 0 to $T$, the drift term decreases exponentially from $\mathbf{x}_0$ to a vanishingly small value, while the noise variance increases exponentially from 0 to 1.

$\gamma >> x_0^j$ ensures that the pixel density drift to a very small value as $i$ approaches $T$. In our experiments for CIFAR10 dataset images of $32 \times 32$ resolution (Krizhevsky, 2009), we fixed $T = 200$ and $\gamma = 20$. For CelebA dataset images of $128 \times 128$ resolution (Liu et al., 2015) , we fixed these values to 500 and 50 respectively. Note that, we opted to keep a 1:10 ratio between $\gamma$ and T. While heuristically chosen, they were subjected to a thorough experimental validation over toy examples. Additionally, due to lower information content in lower resolution images, as in CIFAR10 dataset, the forward diffusion tunes into noise at a much faster rate in comparison to when using higher resolution CelebA images.

To analyze the time trajectory of our diffusion model, we substitute discrete ratio $i/T$ with a continuous variable $t \in [0, 1]$, by letting $T \to \infty$, the discrete time-step Eqn. 9 becomes a continuous time one:

$$\mathbf{x}_t = \sqrt{\overline{\alpha}_t} \odot \mathbf{x}_0 + \sqrt{1 - \overline{\alpha}_t} \odot \tilde{\epsilon}_t, \ \overline{\alpha}_t = e^{-\gamma \, t \mathbf{x}_0 / T}, \ t \in [0, 1]. \tag{10}$$

Using Eqn. 10, we can calculate the time differentiable SNR at pixel $x_0^j$ as:

$$\text{SNR}(j, t) = \frac{(x_0^j)^2}{e^{\gamma \, t \, x_0^j} - 1} \tag{11}$$

Form Eqn. 11, it is clear that the SNR of any pixel decreases exponentially with time.

### 3.2.2 PROPOSITION 1

For our diffusion starting from clean pixels, $x_0^j < x_0^k$, there exists a $t_\delta$,

$$\frac{d \, \text{SNR}(k, t)}{dt} < \frac{d \, \text{SNR}(j, t)}{dt} < 0, \ \forall \, t \in [0, t_\delta] \tag{12}$$

The proof of this proposition can be found in Appendix A. Thus, our choice of $\alpha_i$ makes pixels with high values experience faster SNR reduction.

### 3.2.3 PROPOSITION 2

For the diffusion model defined as per Eqn - 10, the expected trajectory is $\chi_N(t) = \mathbb{E}[\mathbf{x}_t] = \mathbf{x}_0 \odot e^{-\frac{\gamma \mathbf{x}_0 t}{2}}$. Choosing $\gamma x_0^j > at, \ \forall \, j \in \{1, \cdots, d\}, \ t \in (0, 1]$, ensures

$$|\frac{\chi_C(t)}{dt}| < |\frac{\chi_N(t)}{dt}| \tag{13}$$

where $\chi_C(t) = \mathbb{E}[\mathbf{x}_t] = \mathbf{x}_0 e^{-\frac{at^2}{2}}$ is the expected trajectory of a linearly varying conventional DDPM schedule of the form, $\beta_t = at, \ t \in [0, 1]$.

The proof of proposition 2 can be found in Appendix B. Thus, by carefully choosing $\gamma$, our model achieves faster convergence in comparison to conventional DDPM model.

Fig. 2 demonstrates that the mean and variance of our new method converge to target values (0 and 1 respectively) much faster than the conventional diffusion model. Here the total number steps is kept constant to 500 steps. The $\beta_i$ for conventional DDPM vary linearly from $10^{-4}$ to 0.02. The example image is a sample from the CelebA dataset that has a resolution of $128 \times 128$. The means and variances calculated in this experiment are empirical ones. We rely on the law of large numbers that as the number of pixels in the image is sufficiently large (specifically $128 \times 128 = 16,384$), the empirical statistics match theoretical statistics with high probability. Fig. 2 demonstrates that the empirical mean and variance of our new method converge to target values (0 and 1 respectively) much faster than the conventional DDPM model at 500 steps for an image from the CelebA dataset. The $\beta_i$ for DDPM varies linearly from $10^{-4}$ to 0.02. We have assumed that, as the number of pixels in the image is large enough (specifically $128 \times 128 = 16,384$), the empirical statistics would match theoretical statistics with high probability.

At inference time the reverse diffusion process requires the knowledge of the scheduling parameters which are themselves dependent on $\mathbf{x}_\delta = e^{-\gamma \mathbf{x}_0}$. At first glance, this appears to be a counter-intuitive task as acquiring $\mathbf{x}_0$ through a stochastic trajectory seems to require knowledge of $\mathbf{x}_0$ itself. To avoid this dilemma, we take advantage of the feature of $\mathbf{x}_\delta$ that provides only structural information with less fine details, or an approximation of it, to provide some prior information of the image structure. Consequently, we exploit a VAE, a less complex denoising based method to estimate $G_\theta(\mathbf{x}_i, i) = \hat{\mathbf{x}}_\delta \approx \mathbf{x}_\delta$

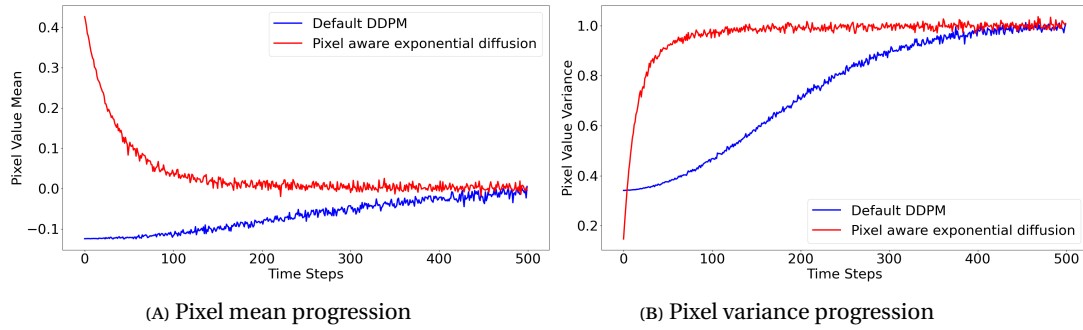

(A) Pixel mean progression

(B) Pixel variance progression

FIGURE 2: Comparison of pixels mean (left) and variance (right) progression in forward diffusion trajectory of a single color channel (red color) over time of the conventional DDPM (blue) vs. our model (red).

from a noisy image $\mathbf{x}_i$. The requirement on parameter complexity of $G_\theta(\mathbf{x}_i, i)$ can be kept low due to $\mathbf{x}_\delta$ lacks finer details (unlike $\mathbf{x}_0$) as a result of exponentiation and the large value of $\gamma$. The image scale, $\mathbf{x}_\delta$ and $\gamma$ are independent of time-steps. Once $\mathbf{x}_\delta$ is estimated, the approximations of the factors $\alpha_i^j$ and $\overline{\alpha}_i^j$ can also be readily calculated to further recover $\mathbf{x}_0$.

Fig. 3 shows, using an example from CelebA dataset, the comparison of the real image, $\mathbf{x}_0$ vs $\hat{\mathbf{x}}_\delta$ generated by $G_\theta(\mathbf{x}_T, T)$. The second image also shows the real image scale, $\mathbf{x}_\delta$. To also verify that it is easier to estimate $\mathbf{x}_\delta$ instead of $\mathbf{x}_0$ from $\mathbf{x}_T$, we also trained a VAE with same architecture and size as $G_\theta(.)$. The outputs from this VAE had many spurious artifacts with very poor generation quality. Example outputs of this VAE are provided in the Appendix F. Similarly, the figures in Appendix E show comparison of real and estimated $\overline{\alpha}_i$. The structural similarity (SSIM) index between real and estimated images was found to be in the range of 0.86 to 0.99 for both $\mathbf{x}_\delta$ and $\overline{\alpha}_i$ (the higher the better, with 1 signifying perfect similarity) across different values of $i$. Appendix G shows progression of our diffusion process compared with conventional DDPM process showing a visual evidence our faster convergence in both forward as well as reverse direction.

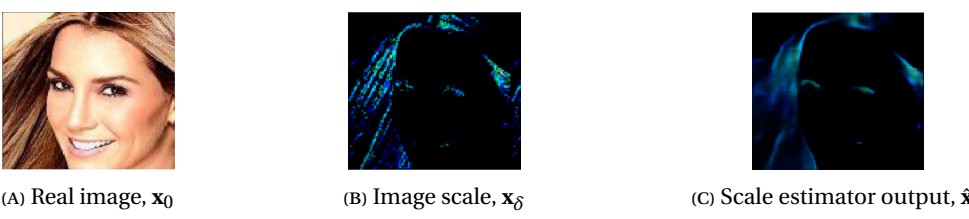

(A) Real image, $\mathbf{x}_0$

(B) Image scale, $\mathbf{x}_\delta$

(C) Scale estimator output, $\hat{\mathbf{x}}_\delta$

FIGURE 3: Illustration of real image vs Image Scale vs Denoised output from VAE.

Details about the architecture of the VAE are in Appendix C. This forward process design accelerates the reverse diffusion process via two sources:

1. Acceleration due to shortened forward trajectory as shown in Proposition 3.2.3 reducing the diffusion process steps by 50-75%.

2. The image scale approximation acting as a regularizer affords a simultaneous parallel generation of reverse diffusion steps avoiding multiple iterative runs of the trained parallel diffusion model. This results in overall reduction of generation time at least by a factor of 4. This is explained in the next sub-section.

### 3.3 Reverse-time diffusion modeling

To proceed with a reverse-time diffusion we estimate $\hat{\mathbf{x}}_\delta$, $\beta_i$, $\alpha_i$ and $\overline{\alpha}_i$ using the VAE, $G_\theta(\mathbf{x}_i, i)$, as discussed in the previous subsection. In conventional designs the same trained network is used iteratively to generate the reverse trajectory sample at a particular time-step by using the generated sample for previous time-step as input. This provides evidence that the same architecture has a sufficient capacity to process the semantic information hidden in the noisy image at any time-step. Theoretically, based on the Universal Approximation Theorem, the conventional GDMs can also try to generate all the steps parallelly by relying on a prohibitively large neural network much too difficult to train.

While conventional GDMs learn to approximate the posterior distribution, $q(\mathbf{x}_{i-1}|\mathbf{x}_i, \mathbf{x}_0)$ as $p_\theta(\mathbf{x}_{i-1}|\mathbf{x}_i)$, our reverse diffusion model on the other hand approximates it using $p_\phi(\mathbf{x}_{i-1}|\mathbf{x}_i, \hat{\mathbf{x}}_\delta)$. Thus, the scale estimator autoencoder, $G_\theta(\mathbf{x}_i, i)$ is central to the proposed model. It generates the image scale approximation as an informed prior regularizing our parallelized reverse diffusion model. The overall architecture of our parallelized reverse diffusion model using a U-net architecture, is shown in Fig.-4 in Appenix C. In adapting it to our proposed methodology, the following modifications are in order:

1. We also fuse (by addition to feature maps) $\hat{\mathbf{x}}_\delta$ predicted from the image scale autoencoder $G_\theta(x_i, i)$.

2. We modify the structure of the last layer of the whole model to predict the additive noise,$\mathbf{Z}^j$ for all the preceding time-steps ($j \in \{i-1, \cdots, 1\}$) in different channels of the last layer. While the one time complexity of our model is higher than existing competing models, unlike the latter, only a single execution is required of the trained model to obtain a clean image $\mathbf{x}_0$. This thus results in an overall reduction in sample generation time.

Please refer to Appendix C for more details. The model $R_\phi(\mathbf{x}_i, \hat{\mathbf{x}_\delta}, i)$ (with trainable parameters, $\phi$) receives a noisy image $\mathbf{x}_i$, its predicted scale $\hat{\mathbf{x}_\delta}$ and time-step information $i$ as input and predicts additive noise for all the steps of the forward diffusion in parallel. These predictions can then be used to generate the reverse trajectory using Eqn.-6 with $\epsilon_\phi(x_j, j)$ replaced by predictions, $\mathbf{Z}^j$, $j \in \{1, \cdots, T\}$ of the model.

The last layer of the whole network has $T$ feature maps. The $j^{th}$ feature map, $\mathbf{Z}^j$, $j \in \{1, \cdots, T\}$, is:

$$\mathbf{Z}^j = \mathcal{M}_j \odot \mathcal{G}_j\Big(\mathcal{Z}_{\mathcal{D},K}, \mathcal{P}(i), \mathcal{H}(\hat{\mathbf{x}}_\delta); \phi_j\Big), \tag{14}$$

where, $\mathcal{Z}_{\mathcal{D},K}$ is the output of the last decoder, with $K$ being the total number of decoders. $\phi_j \subset \phi$. $\mathcal{M}_j$ is a channel mask which is all 1's if $j < i$, and otherwise all 0's. This ensures that only predictions for time-steps preceding $i$ are made. $\mathcal{G}_j(.)$ is a feature map implemented using a small neural network. The fusion of hidden information in $\mathbf{x}_\delta$, estimated as $\hat{\mathbf{x}}_\delta$ allows us to reduce the parameter complexity of $\mathcal{G}_j(.)$. $\mathcal{P}(i)$ is a non-linear mapping of the input time-step $i$ with the same dimensions as a single channel of the decoder output, $\mathcal{Z}_{\mathcal{D},K}$. $\mathcal{H}(\hat{\mathbf{x}}_\delta)$ is a non-linear mapping of $\hat{\mathbf{x}}_\delta$. The parallel prediction of noise outputs for all the steps (per Eqn.-14 ) underlines the differences with a conventional U-Net based design. So, for an RGB image, while conventional U-Net models have a 3-channel output, our model has a 3 x 500 channels output for a 500 time-steps trajectory. Additionally, just like fusion of time-step embedding in conventional models, image structure information in the form of $\mathbf{x}_\delta$ is also fused inside the U-Net. All $T$ optimization objective functions are similar to those used in conventional DDPM (Ho et al., 2020):

$$L(\phi, j) = \mathbb{E}_{i, x_i}[|||Z^j - \tilde{\epsilon}_j||_2^2], \ j \in \{1, \cdots, T\} \tag{15}$$

For $j > i$, $L(\phi, j)$ is fixed to 0 as a consequence of the same argument of using the mask $\mathcal{M}_j$. $L(\phi, j)$ for different values of $j$ are optimized in parallel. The parameters of the common network backbone are thus:$\phi_b = \{\phi_l | \phi_l \in \phi, \phi_l \notin \phi_j, \forall j\}$ are trained by all the $L(\phi, j)$, while the parameters $\phi_j$ of a particular $\mathcal{G}_j(.)$ are trained only by its particular loss function $L(\phi, j)$.

The procedure for generating the final clean image is shown in Algorithm 1. It is similar to the one used by (Ho et al., 2020), the difference being that the scheduling parameters are calculated from $\hat{\mathbf{x}}_\delta$ and only a single forward pass through the model $R_\phi(.)$ is required to predict all denoising terms $\epsilon_\phi(., j)$ in the Eqn.-4, as they are available in parallel in the form of $Z^j$.

## 4 EXPERIMENTS AND RESULTS

The models were trained on Cifar10 and CelebA datasets for fair comparison with other models. The images were first normalized to the range $[\epsilon, 1]$. Note that a small $\epsilon = 4 \times 10^{-3}$ is added to all $x_0^j$ to ensure their values are greater than 0 so that $\overline{\alpha}_i$ vary with $i$. Time-step inputs to the modified U-net for the reverse diffusion model were encoded using sinusoidal positional embedding (Vaswani et al., 2017).

Appendix H shows some generated examples for CIFAR10 and CelebA datasets. We compared our algorithm to DDPM (Ho et al., 2020), a discrete time model like ours, the Stochastic Differential Equation (SDE) based continuous model introduced by Song et al (Song et al., 2021b), DDIM (Song et al., 2021a), an accelerated modification of DDPM and DPM Solver applied over vanilla DDPM(Lu et al., 2023). Tables 1 and 2 show image generation performance of the these models on CIFAR10 and CelebA datasets respectively in terms of number of trainable parameters, FID scores and execution time. Training and inference was done on a single NVIDIA Tesla V100 SXM2 32 GB GPU. Most research efforts, such as (Song et al., 2023; Lu et al., 2022; Salimans & Ho, 2022; Chen, 2023; Chen et al., 2024), focus on fast sampling using fast ODE solvers applied to the backward diffusion of SDE based model. A continuous time version of our model will appear in a future paper as it is out of scope of the present paper due to space limit. These fast solvers will be compatible and can be applicable to our continuous time model as well, providing further acceleration.

While the image quality of our model is competitive, its execution time is at least 4 times lesser than that of vanilla DDPM (our 0.3 second against their 1.23 seconds in case of CIFAR10) and comparable to DDIM even when more time-steps are used (our 500 vs their 100 steps). The SDE based model effectively requires orders of magnitude more steps as a result of correction required due to MCMC subsampling. Our model on the other hand requires only 200 time-steps and 500 time-steps long trajectory to provide comparable performance, and even these steps of the trajectory are generated in a single run of the model. In general, generation quality is proportional to the number of time-steps in the trajectory. This is evident in Tables-1 and 2 from the decreasing FID scores in the case of DDIM and DPM Solvers with increasing number of time-steps. Our models produce lower FID scores in comparison to DDIM models with 1000 steps. DDIM models with fewer steps produce higher FID scores. Same holds true for DPM Solver based outputs also. Our model has more trainable parameters because of the added scale VAE, $G_\theta(.)$ and the multiple parallel out channels, $\mathbf{Z}^i$ in $R_\phi(.)$. Specifically, in the case of model trained for CIFAR10 the 71.5 million parameters comprise 30.3 million parameters of VAE and 41.2 million parameters of the revers diffusion model. In the case of the model trained for the CelebA dataset, these numbers were 61.3 million and 84.2 million respectively. Consequently, the slightly increased parameter complexity in our models is compensated by just a single forward pass required by our model.

## 5 LIMITATIONS AND FUTURE WORK

In addition to keeping T smaller than that of conventional DM, our pixel-aware forward diffusion approach yields an overall accuracy and time advantage over conventional models. Reducing the number of time-steps $T$ of the diffusion process sets an upper bound on the choice of $\gamma$. Larger $\gamma$ results in faster convergence to standard normal distribution in the forward direction and provides image scale without detailed features for reconstruction. Using a value greater than $T$ results in an

| Model | #Param(M) | #Steps | FID | Time(sec) |
|---|---|---|---|---|
| DDPM | 35.7 | 1000 | 3.28 | 1.26 |
| SDE based | 31.4 | 1000 | 2.99 | 47.67 |
| DDIM | 35.7 | 10 | 13.36 | 0.03 |
| DDIM | 35.7 | 100 | 4.16 | 0.33 |
| DDIM | 35.7 | 1000 | 4.04 | 3.22 |
| DPM Discrete Solver | 35.7 | 10 | 5.37 | 0.02 |
| DPM Discrete Solver | 35.7 | 34 | 4.16 | 0.05 |
| DPM Discrete Solver | 35.7 | 100 | 3.94 | 0.15 |
| DPM Discrete Solver | 35.7 | 200 | 3.77 | 0.31 |
| DPM Discrete Solver | 35.7 | 500 | 3.41 | 0.76 |
| Our Model | 71.5 | 200 | 3.15 | 0.3 |

TABLE 1: CIFAR10 Generative Performance

| Model | #Param(M) | #Steps | FID | Time(sec) |
|---|---|---|---|---|
| DDPM | 78.7 | 1000 | 3.51 | 10.19 |
| SDE based | 65.6 | 1000 | 3.20 | 246.69 |
| DDIM | 78.7 | 10 | 17.33 | 0.53 |
| DDIM | 78.7 | 100 | 6.53 | 5.55 |
| DDIM | 78.7 | 1000 | 3.51 | 48.44 |
| DPM Discrete Solver | 78.7 | 10 | 4.85 | 0.04 |
| DPM Discrete Solver | 78.7 | 34 | 4.53 | 0.12 |
| DPM Discrete Solver | 78.7 | 100 | 4.52 | 0.33 |
| DPM Discrete Solver | 78.7 | 500 | 3.79 | 1.48 |
| Our Model | 145.5 | 500 | 3.25 | 1.3 |

TABLE 2: CelebA Generative Performance

exceedingly fast decay in the forward direction which breaks the Markovian condition, while using a smaller value results in non-convergence of some lower valued-pixels to standard normal distribution. Lifting the hyperparameter nature of $\gamma$ is part of our future work, and our plan is for a systematic choice by jointly considering the data/pixels distribution to optimize some associated energy functional while ensuring the existence of a stationary distribution as is commonly the case for a Markov chain. Additional experiments with much more diverse datasets are also planned for a more profound understanding of a universal selection of parameters.

A particularly interesting angle about this stochastic diffusion is to investigate the performance of our proposed diffusion by exploring the internal mechanics of the encoder and decoder in the context of diffusion processes. Our model learns the reverse diffusion process conditional on the global image structure (via $\mathbf{x}_\delta$). A future challenge is to discover the extent of control over a diffusion process for a system of interacting particles (Bao & Krim, 2004; Krim & Bao, 1999) instead of parallel independent forward diffusions as is the norm in current models. Given the asymmetry of information processing between the encoder and decoder implementation of the U-Net model uncovered in (Li et al., 2023), it would be interesting to study what kind of semantic information is learned by the two sub-components of the model. Finally, we also intend to study how our approach works on transformer based models to improve performance.

## 6    CONCLUSION

In this paper, we have introduced a novel forward diffusion model that significantly improves upon the limitations of conventional models in terms of convergence speed and computational efficiency. By leveraging the microscopic structure of clean images to learn the drift and diffusion coefficients, our model degrades the Signal to Noise Ratio much faster than traditional approaches. Drawing inspiration from the water pouring algorithm, we implemented a pixel-based scheduling strategy that optimizes forward diffusion by considering the initial values of individual pixels. This method achieves isotropic Gaussian distribution across the pixels more efficiently than conventional, pixel-agnostic diffusion methods.

Furthermore, we utilized an autoencoder to learn a comprehensive diffusion schedule. The learned knowledge of the global structure of the clean image inspired us to develop a reverse-time data driven diffusion model to generate the entire reverse-time diffusion trajectory in one step. This approach not only maintained image quality but also accelerated the reverse-time diffusion process by up to 10 times compared to existing models. Our findings demonstrate that addressing the inefficiencies of universal diffusion in generative models through a detailed, pixel-focused approach can lead to substantial improvements in performance, paving the way for more effective and efficient generative image modeling.

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

## A   PROOF THAT EXPONENTIAL PIXEL VALUE BASED DIFFUSION FOLLOWS WATER POURING ALGORITHM

In this section, we will show that in each step higher variance noise is being added to a higher value pixel in comparison to a lower value pixel.

SNR of pixel $j$ at time $t$ is calculated as

$$\text{SNR}(j, t) = \frac{(x_0^j)^2}{e^{\gamma\, t\, x_0^j} - 1} \tag{16}$$

Thus, the rate of change of SNR over time $t$ is

$$\frac{d\,\text{SNR}(j, t)}{dt} = -\frac{\gamma (x_0^j)^3}{(e^{\gamma\, t\, x_0^j} - 1)^2} \tag{17}$$

From the Taylor series

$$e^x = 1 + x + \frac{x^2}{2!} + \cdots$$

we can show that for any small value $x > 0$, the following inequality is true.

$$0 < xe^{x/2} < e^x - 1 < xe^x$$

This leads to

$$0 < \frac{1}{xe^x} < \frac{1}{(e^x - 1)} < \frac{1}{xe^{x/2}}$$

Applying above inequality to our case, we can see that SNR at time $t$ can be expressed as

$$0 < \frac{x_0^j}{(\gamma\, t)e^{\gamma\, t\, x_0^j}} < \frac{(x_0^j)^2}{e^{\gamma\, t\, x_0^j} - 1} < \frac{x_0^j}{(\gamma\, t)e^{\gamma\, t\, x_0^j/2}} \tag{18}$$

And the magnitude of rate of change of SNR with time, $t$ can be expressed as

$$0 < \frac{x_0^j}{\gamma\, t^2\, e^{2\gamma\, t\, x_0^j}} < \frac{\gamma (x_0^j)^3}{(e^{\gamma\, t\, x_0^j} - 1)^2} < \frac{x_0^j}{\gamma\, t^2\, e^{\gamma\, t\, x_0^j}} \tag{19}$$

From Eqns. 18 and 19 we can see that for $t$ belonging to a small interval, $t \in [0, t_\delta]$, both the SNR as well as the magnitude of its rate of change of SNR at pixel $x_0^j$ are very high and it quickly approaches 0. In this small interval $[0, t_\delta]$, the SNR decays at the rate of $t^{-2}$. The decay rate is proportional to $x_0^j$ also. Therefore, when $0 < x_0^j < x_0^k$, in the interval $t \in [0, t_\delta]$, we have

$$\left|\frac{d\,\text{SNR}(j, t)}{dt}\right| < \left|\frac{d\,\text{SNR}(k, t)}{dt}\right|$$

## B   PROOF OF FASTER CONVERGENCE OF OUR FORWARD DIFFUSION PROCESS

For any discrete diffusion process satisfying, Eqn. 2, the corresponding continuous SDE is

$$d\mathbf{x}_t = -\frac{\beta_t}{2}\mathbf{x}_t dt + \sqrt{\beta_t}d\mathbf{w}, \;\; t \in (0, 1) \tag{20}$$

From theory of Ito Calculus (Särkkä & Solin, 2019; Oksendal, 2000; Lara, 2006), we have

$$x_t - x_0 - \int_0^t \mathbf{f}(\mathbf{x}_s, s) ds = x_t - x_0 - \int_0^t -\frac{\beta_s}{2} \mathbf{x}_s ds$$

is a martingale (for definition see (Oksendal, 2000)). Thus we can calculate rates of change of mean of the image at $t$ as:

$$\frac{d \mathbb{E}[x_t]}{dt} = \mathbb{E}[\mathbf{f}(\mathbf{x}_t, t)] = -\frac{\beta_t}{2} \mathbb{E}[x_t] \tag{21}$$

This leads to

$$\mathbb{E}[x_t] = x_0 e^{-\int_0^t \frac{\beta_s}{2} ds}$$

To focus on analyzing the algorithm, we simply let $\beta_t = at$ (a linear function of time). Hence the expected trajectory of the DDPM follows the exponential decay of the form

$$E[x_t] = x_0 e^{-\frac{at^2}{2}}. \tag{22}$$

We now compare the expected trajectory of our model with the trajectory in Eqn 22

The continuous SDE corresponding to our pixel-wise diffusion given in 3.2.3 is:

$$dx_t^j = -\frac{\gamma x_0^j}{2} x_t^j dt + \sqrt{\gamma x_0^j} dw^j, \tag{23}$$

This SDE can be derived from the discrete iterative forward diffusion equation of our process using the same approach as described in (Song et al., 2021b).

In our case, the rate of change of mean turns out to be:

$$\frac{d \mathbb{E}[x_t^j]}{dt} = -\frac{\gamma x_0^j}{2} \mathbb{E}[x_t^j] \tag{24}$$

This leads to

$$\mathbb{E}[\mathbf{x}_t] = \mathbf{x}_0 \odot e^{-\frac{\gamma \mathbf{x}_0 t}{2}} \tag{25}$$

By further choosing $\gamma$ in such a way that $\gamma x_0^j > at, \ \forall \ j \in \{1, \cdots, d\}, \ t \in (0, 1]$, we can ensure that the trajectory in Eqn 25 decays faster than the DDPM trajectory in Eqn 22.

This demonstrates that our diffusion accounts for the pixel value in deciding the rate of decay in the forward process, which in turn requires a selection of a specific value of $\gamma$ as described in Section 3.2.1 to ensure effective pixel-wise diffusion.

The drift term of each pixel diffusion can be further generalized to $-\Gamma(x_0^j, t) x_0^j$, where $\Gamma(x_0^j, t)$ is a positive function of $t$ and $x_0^j$, monotonic in $t$. Similarly, Over all the pixels, we can have a vector valued function $\mathbf{\Gamma}(\mathbf{x}_0, t) \in \mathbb{R}^d$.

For comparison with conventional DDPM, one can select $\mathbf{\Gamma}(\mathbf{x}_0, t) = \gamma \frac{\beta_t}{2} \mathbf{x}_0$, where $\beta_t$ is the same parameter used in conventional DDPM. This makes the diffusion schedule pixel-dependent. If $\beta_t = at$, we will have

$$\mathbb{E}[\mathbf{x}_t] = \mathbf{x}_0 \odot e^{-\frac{\gamma a \mathbf{x}_0 t^2}{2}} \tag{26}$$

This demonstrates the acceleration our pixel-wise modulated diffusion has over the DDPM algorithm.

In the case of a single pixel, we have an exponential function that we can illustrate using the pixel value. We have also compared the conventional diffusion models with our design by conducting experiments with toy examples, where noise added in the forward direction is completely known beforehand. Specifically, we conducted reverse trajectory generation experiments where the entire forward direction trajectory (i.e., the noise samples $\mathbf{Z}$ in Step-5 of the Sampling algorithm-1) was known beforehand. In Sampling algorithm-1 the scale estimate $\hat{\mathbf{x}}_\delta$ was replaced with actual scale value $\mathbf{x}_\delta$ and we checked the MSE between generated reverse direction image, and the real image $\mathbf{x}_0$. Appropriate value of $\gamma$ was chosen based on these experiments.

## C  MODEL ARCHITECTURE AND OTHER DETAILS

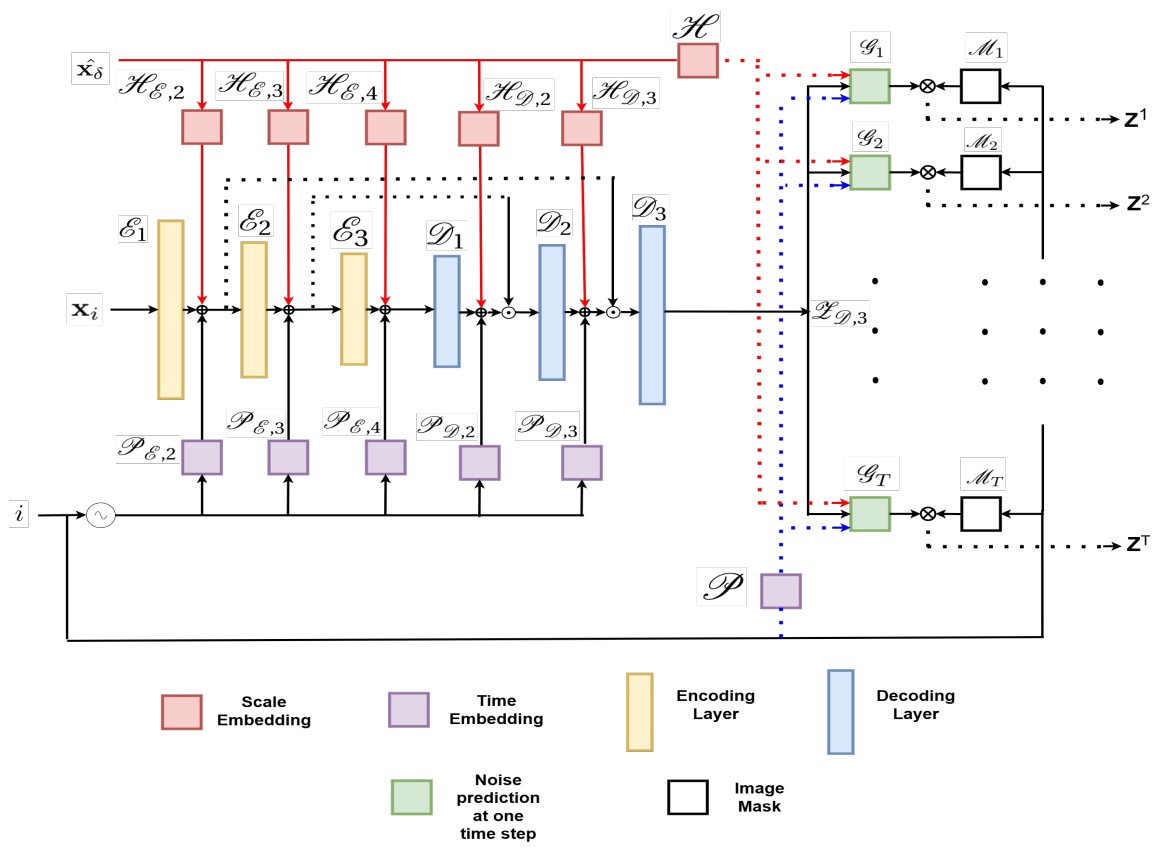

FIGURE 4: Reverse diffusion model architecture

As autoencoder training involved learning simpler image structures, the depth of the network is kept much lower. Each autoencoder encoder/decoder layer is composed of a single residual convolutional block along with self attention of $16 \times 16$ context size followed by a downsampling/upsampling by an order of 2. Group normalization is applied to the input of each convolutional layer. The time-steps are encoded using Sinusoidal positional embedding with encoding vector size chosen as 128. The bottleneck

layer outputs composed of feature maps of $4 \times 4$ resolution are flattened and projected to a larger size latent space to ensure generation diversity.

Fig.-4 shows the architecture of our parallelized reverse diffusion model. Our reverse diffusion model is based on U-Net architecture. The same time encoding employed by the autoencoder was also used by the reverse diffusion models. Salient aspects of our reverse diffusion model are:

1. Like conventional U-Net, we have multiple layers of downsampling encoders, $\mathscr{E}_k$ followed by multiple upsampling decoders, $\mathscr{D}_k$, $k \in \{1, \cdots, K\}$. Each encoder/decoder layer consists of 2 residual convolutional blocks with self attention.

2. At the input of any decoder/encoder, $\mathscr{D}_k/\mathscr{E}_k$ of layer $k$, information of the input time-step information, $i$ and image scale estimate, $\hat{\mathbf{x}}_\delta$ are fused by adding their learnable non-linear mappings, $\mathscr{P}_k(.)$ and $\mathscr{H}_k(.)$ respectively.

3. The last decoder layer, $\mathscr{D}_K$ has a 512 channels output, $\mathscr{Z}_{\mathscr{D},K}$

The final layer involves parallel sub-networks, $\mathscr{G}_j(.)$, $j \in \{1, \cdots, T\}$ to model various time-step noise predictions. Each sub-network is modeled as 2 successive residual blocks with self attention with the number of output channels being same as the input image (i.e. 3 in case of RGB images) generated using Eqn.-14. Adam optimizer is used for backpropagating through all the networks. The learning rates used for CIFAR10 and CelebA database are respectively $10^{-4}$ an $2 \times 10^{-4}$ respectively, with training being done over 1.5 M iterations. The batch sizes for the two datasets are, respectively, 128 and 16. Finally, the number of time-steps, $T$ of the diffusion process are fixed to 200 and 500 respectively. Training and inference was done on a single NVIDIA Tesla V100 SXM2 32 GB GPU. The same GPU was used to compare execution time of conventional models.

## D  ALGORITHM FOR SAMPLING FROM THE TRAINED REVERSE DIFFUSION MODEL

---

ALGORITHM 1: Sampling algorithm

---

**Require:**  Pre-trained scale autoencoder $G_\theta(.)$ and reverse diffusion model, $R_\phi(.)$.
**Input:**  Noisy image $\mathbf{x}_i$ and time-step $i$ of the forward diffusion
**Output:**  Clean image $\hat{\mathbf{x}}_0$

1: Scale estimate: $\hat{\mathbf{x}}_\delta = G_\theta(\mathbf{x}_i, i)$

2: $\alpha_j = \exp\left\{(\frac{1}{T}\log(\hat{\mathbf{x}}_\delta))\right\}, \ \forall j \in 1, \cdots, T$

3: $\overline{\alpha}_j = \exp\left\{(\frac{j}{T}\log(\hat{\mathbf{x}}_\delta))\right\}, \ \forall j \in 1, \cdots, T$

4: $\tilde{\beta}_j = \dfrac{1 - \overline{\alpha}_{j-1}}{1 - \overline{\alpha}_j}(1 - \alpha_j), \ \forall j \in 1, \cdots, T$

5: Reverse diffusion noise predictions: $\mathbf{Z} = \{\mathbf{Z}^1, \cdots, \mathbf{Z}^T\} = R_\phi(\mathbf{x}_i, \hat{\mathbf{x}}_\delta, i)$

6: **Initialization** $j = i$, $\hat{\mathbf{x}}_j = \mathbf{x}_i$

7: **while** $j > 0$ **do**

8:    $\epsilon \sim \mathcal{N}(\mathbf{0}, \mathbf{I})$

9:    $\hat{\mathbf{x}}_{j-1} = \dfrac{1}{\sqrt{\alpha_j}} \odot \left(\hat{\mathbf{x}}_j - \dfrac{1 - \alpha_j}{\sqrt{1 - \overline{\alpha}_j}} \odot \mathbf{Z}^j\right) + \sqrt{\tilde{\beta}_j} \odot \epsilon$

10:    $j = j - 1$

11: **end while**

12: **return** $\hat{\mathbf{x}}_0$

---

## E    REAL AND LEARNED SCHEDULING PARAMETERS COMPARISONS

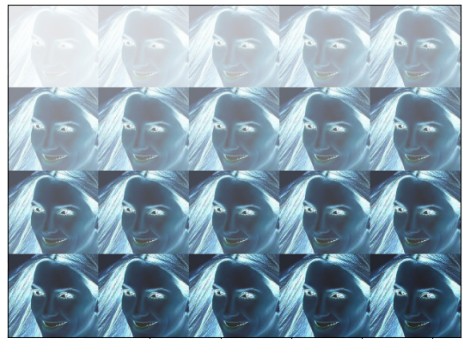
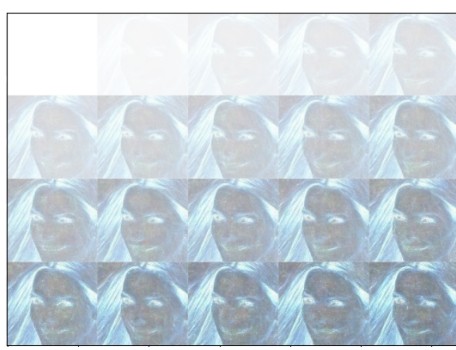

(A) Real $\overline{\alpha}_i$

(B) Estimated $\overline{\alpha}_i$

FIGURE 5: Real $\overline{\alpha}_i$ and estimated $\overline{\alpha}_i$ (calculated from the image scale estimate, $\hat{\mathbf{x}}_\delta$ for i ranging from 1 to 20.

## F    EXAMPLE OF DIRECT DENOISING ATTEMPT

To prove our design choice of using the VAE, $G_\theta(\mathbf{x}_i, i)$ to estimate image scale first, we also trained another VAE with same architecture and number of parameters as our Scale Estimator, $G_\theta(\mathbf{x}_i, i)$ to directly denoise the noisy image $\mathbf{x}_i$. Fig.-6 shows example outputs from this VAE. As is evident in the figure, the outputs have various unnecessary artifacts. In comparison, as is evident from Fig.-3, our scale estimator VAE doesn't have such spurious artifacts. $\hat{\mathbf{x}}_\delta$ works as an informational prior in our reverse generative model, without unnecessary spurious information, thus reducing their complexity. $\hat{\mathbf{x}}_\delta$ with only coarse level information about the image structure, is a better-informed prior.

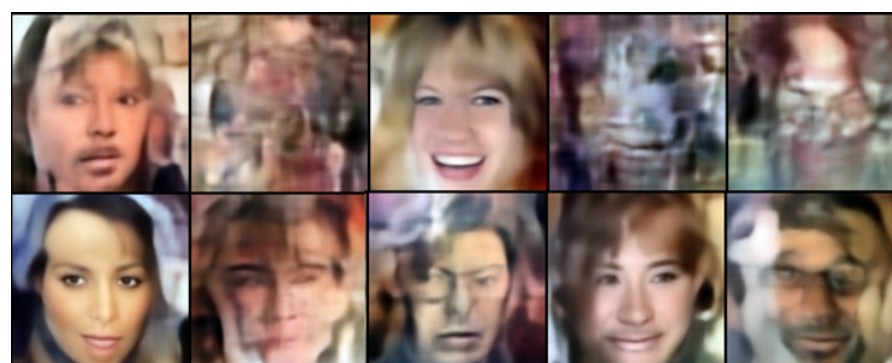

FIGURE 6: Examples of images generated from a denoising VAE with same architecture as our Scale estimator VAE

## G    FORWARD AND REVERSE PROGRESS EXAMPLE

Fig.-7 shows progression of our pixel aware diffusion process of 500 time-steps. For sake of brevity and easier presentation only a subset of 500 time-steps (seleted at every 20th time-step) is shown. Fig.-8 shows conventional DDPM process applied on the same image again over 500 time-steps. As is visible, our process converges faster than conventional DDPM. For more clarity, Fig.-9 and 10 show first 100 steps in the case of forward direction, and last 100 steps in the case of reverse direction of the same processes sub-sampled at every $5^{th}$ time-step.

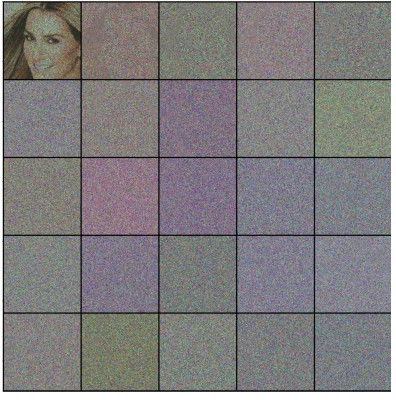
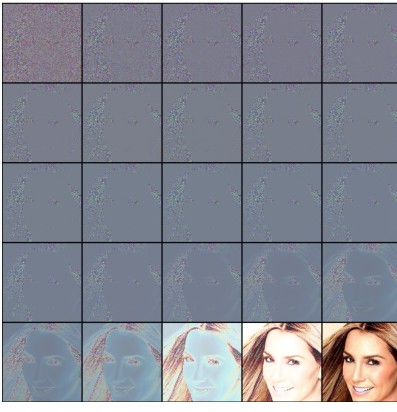

(A) Pixel-aware forward diffusion process.    (B) Pixel-aware reverse diffusion process.

FIGURE 7: Pixel-aware diffusion process progression of 500 time-steps (showing here only a subset of 500 images, selected at every 20th time-step).

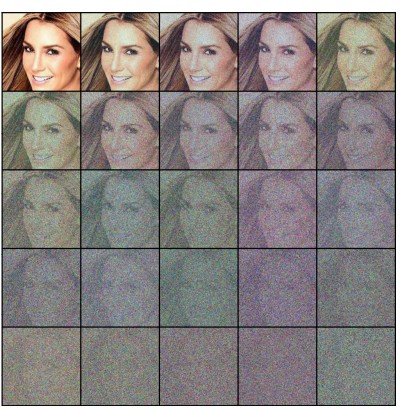
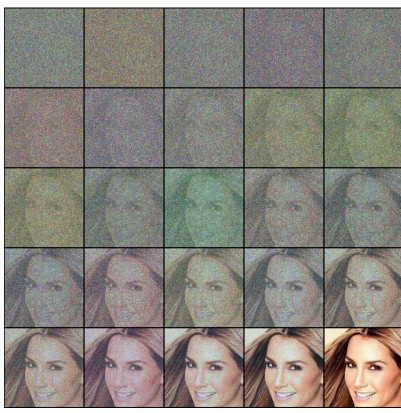

(A) Conventional DDPM forward diffusion process.    (B) Conventional DDPM reverse diffusion process.

FIGURE 8: Conventional DDPM diffusion process progression of 500 time-steps (showing here only a subset of 500 images, selected at every 20th time-step).

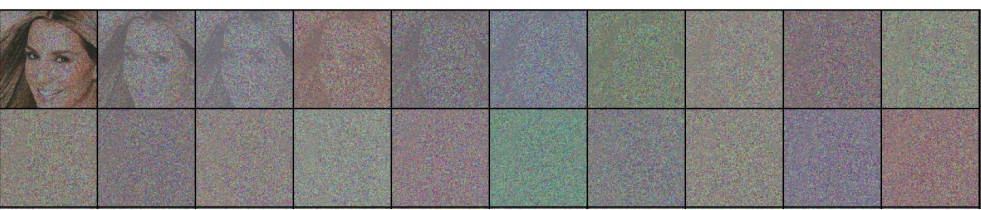

(A) Pixel-aware forward diffusion process.

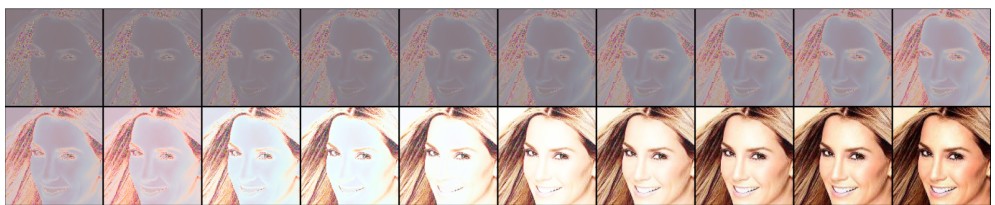

(B) Pixel-aware reverse diffusion process.

FIGURE 9: Pixel-aware diffusion process progression of first 100 time-steps for forward and last 100 time-steps for the reverse direction (showing here only a subset of 100 images, selected at every 5th time-step).

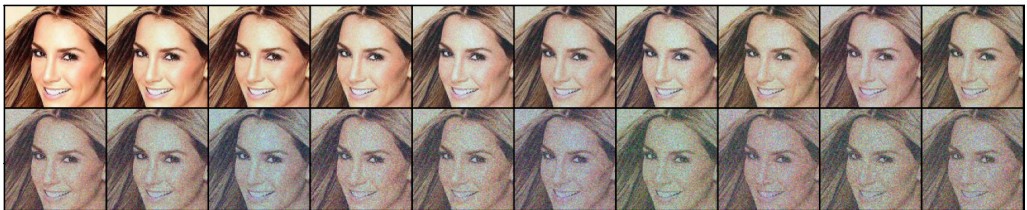

(A) Conventional DDPM forward diffusion process.

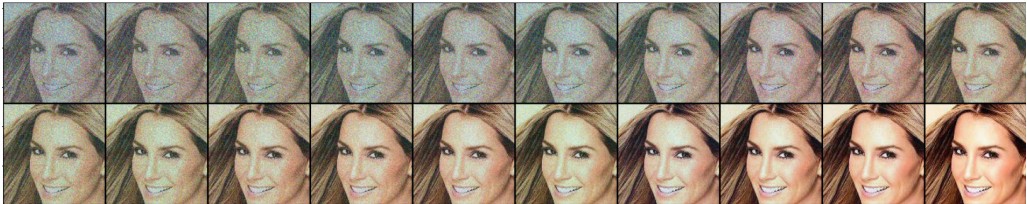

(B) Conventional DDPM reverse diffusion process.

FIGURE 10: Conventional DDPM diffusion process progression of first 100 time-steps for forward and last 100 time-steps for the reverse direction (showing here only a subset of 100 images, selected at every 5th time-step).

# H  GENERATED IMAGE EXAMPLES

FIGURE 11: Cifar10 examples

987
988
989
990
991
992
993
994
995
996
997
998
999
1000
1001
1002
1003
1004
1005
1006
1007
1008
1009
1010
1011
1012
1013
1014
1015
1016
1017
1018
1019
1020
1021
1022
1023
1024
1025
1026
1027
1028
1029
1030
1031
1032
1033

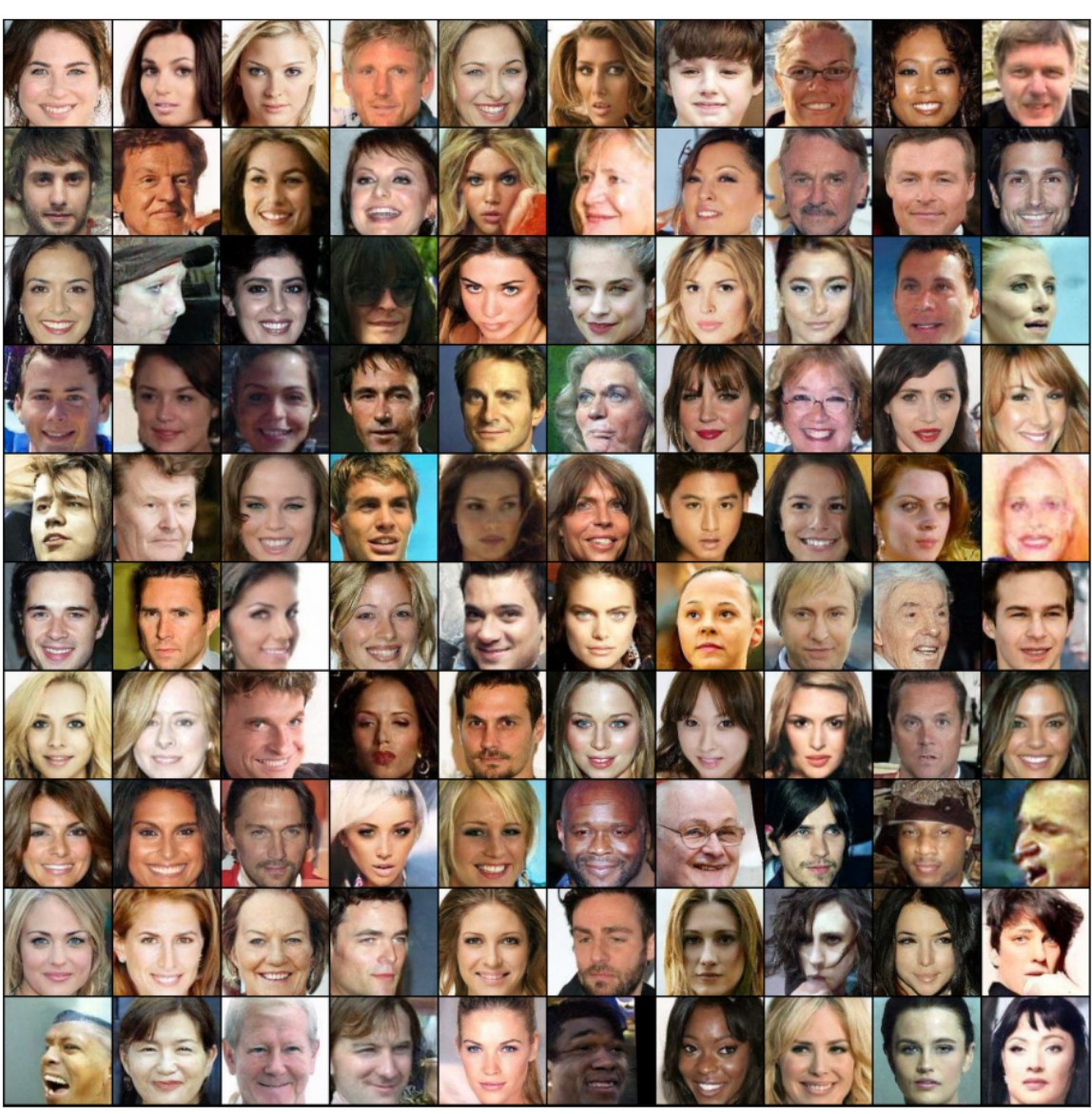

FIGURE 12: CelebA examples

