# OpenReview forum: "Pixel-Aware Accelerated Reverse Diffusion Modeling"
_ICLR.cc/2025/Conference — Submitted to ICLR 2025_

### Official Review · Reviewer_T5eD · 2024-10-17

**Soundness:** 1
**Presentation:** 1
**Contribution:** 1
**Rating:** 3
**Confidence:** 4

**Summary:**

This paper proposes adjusting the diffusion coefficient for each pixel to achieve faster diffusion sampling compared to DDIM. To do this, the authors suggest that different scheduling is required because the SNR varies by pixel (Section 3.1), explain how to implement this (Section 3.2), and propose a modified network structure to accommodate this approach (Section 3.3).

**Strengths:**

I couldn't find any strengths in the manuscript.

**Weaknesses:**

1. **Lack of Recent GDM Studies**: This paper lacks a comprehensive study of recent Generative Diffusion Models (GDMs). While it proposes a new sampler for GDMs, it does not even compare its performance with DPM-Solver, the default sampler for Stable Diffusion. DPM-Solver is only briefly mentioned in L408, but no experiments were conducted, due to *"space limitations"* (L410). This is problematic, as DPM-Solver is a training-free method that is much faster than DDIM and delivers comparable FIDs to DDIM. The proposed method is likely to be outperformed by DPM-Solver. Furthermore, most references throughout the paper are from before 2023, indicating a lack of engagement with the most recent advancements.

2. **Writing Issues**: The writing quality is poor. For example, in Fig. 1, the term "dual" is used, which typically refers to the dual form of a convex optimization problem, making it an inappropriate choice here. Additionally, there are missing captions in Tables 1 and 2, and the citation format is incorrect, as "citep" should have been used in many cases (e.g., L405). Significant figures in Tables 1 and 2 are inconsistent. In L400, the term "dissusion" is misspelled. Figures throughout the paper, especially Figures 2 and 4, have text that is too small to read. Equations (14) and (15) deviate significantly from the standard format in this field, making them difficult to understand. There are many other writing issues throughout the paper.

3. **Unclear Theory**: It is unclear why the SNR is defined as Equation (11). It does not align with the general definition of SNR. In most GDMs, SNR is defined independently of latents (i.e., $x_t$), but in this work, the authors propose making it dependent, which is a significant drawback. To address this, Section 3.3 introduces a new model, G, but as scalability increases, so do the resources required for training and inference (2 $\times$ Params in Tables 1, 2).

**Questions:**

1. In Figure 2, why are the starting points at t=0 different? I believe that they should start with the same value.

---

> ### Author Response · Authors · 2024-11-20
> **Response to reviewer's queries**
>
> We thank Reviewer for their time and comments. We have carefully addressed all questions and concerns below. We will happily provide further clarification if needed.
>
> **Q1** Lack of Recent GDM Studies:
>
> Reply: We believe there is some misunderstanding on the part of Reviewer about our contributions. As noted in our response to other reviewers; we maintain that our approach fundamental changes in the forward diffusion stage which hasn't been addressed by earlier works. Earlier models have only sought to reduce the reverse trajectory by employing subsampling or fast ODE solver based strategies.  We provided an overview of the closely related topics to our work, which we believe will subsequently benefit these backward diffusion solvers to make them more efficient. That being said, we also included a comparison to DPM solver and showed that our processing time in backward diffusion alone is comparable to algorithms including DMP-Solver. Considering that our forward diffusion trajectory has been shortened, our reverse  trajectory is also shortened, and its parallelization  by  our introduction of of the image scale $x_{\delta}$  provides significant run-time reduction.
>
> **Q2** The term "dual" is used, which typically refers to the dual form of a convex optimization problem, making it an inappropriate choice here.
>
> Reply: Point well taken. The ‘dual’ word has various contexts and is used here to refer to dual function space. We have corrected the typo the review pointed out in the revision and added explanation. We respectfully request the reviewer to further clarify their comment on: Significant figures in Tables 1 and 2 are inconsistent for us to address the issue.
>
> **Q3** It is unclear why the SNR is defined as Equation (11). It does not align with the general definition of SNR.
>
> Reply: Noise is added in each step of the forward diffusion process (along with attenuation of the original information content), the SNR does decrease with increasing time. In other words, $SNR(t_2) < SNR(t_1),  \forall t_2 > t_1$. This is true for any diffusion model, including ours, and we use the standard notion of SNR used in Information Sciences.
>
> **Q4** In Figure 2, why are the starting points at t=0 different? I believe that they should start with the same value.
>
> Reply: The starting points are different, because conventional DDPM based models start with clean image, $x_0$ with pixel values normalized in the range [-1,1]. Our model on the other hand, requires the pixel values of $x_0$ to be positive to ensure that all the elements of $\alpha_i$’s are less than 1 (Please refer Eqns-7 and 8). In any case, equality of the starting point of the forward diffusion process is not critical. The aim is to completely remove any useful information at the end of the process (signified by 0 SNR in the end). As is evident in Fig.-2, both conventional as well as our design do achieve 0 SNR (implied by 0 mean and non-zero unit variance) in the end by following different trajectories. In our case, the trajectory is accelerated to achieve 0 SNR (signified by a steeper slope).

---

> > ### Comment · Reviewer_T5eD · 2024-11-25
> >
> > Thanks for your response, for conducting additional experiments, and for updating the manuscript.
> >
> > However, if you ask me, there is still a far way to go to be accepted:
> >
> > **Presentation is not good**: As other reviewers have mentioned, the presentation is lacking. Accelerating diffusion models is a highly popular area of research, yet the paper includes few citations and comparative analyses with prior work in this domain. Additionally, it seems to attempt to claim too many things at once. Section 3 spans from page 4 to page 8, but I feel that the key contributions should have been isolated and emphasized, while the remaining content could have been supported by references to existing studies. To assert novelty, paradoxically, it is crucial to articulate commonalities effectively (for example, it is easier to describe the differences between a mug and a paper cup than between a mug and a headset). The revised version even uses a different font (in my monitor) compared to standard ICLR submissions, which might give the impression of disregarding industry standards and prior studies.
> >
> > **Fails to demonstrate practicality**: Comparison with SoTA models is considered the author’s homework. While DPM-Solvers have been added (thanks), the paper still needs to demonstrate whether the proposed approach can effectively work in conjunction with other acceleration techniques. This is necessary to convince readers that this paper addresses a critical issue at this point in time. For instance, as reviewers aRbR or TBgv mentioned, it would be valuable to show whether the proposed method can be used alongside a Consistency Model or, if the intent is to claim it as a completely different approach, demonstrate that it can generate images as quickly as a Consistency Model. This would make the paper significantly stronger.
> >
> > I would like to maintain my score of 3.

---

> > > ### Author Response · Authors · 2024-11-27
> > >
> > > We profusely thank the reviewer for his/her comments.
> > > We would like to recall and emphasize that the original goal of our paper was to provide a rather different perspective on a problem which has indeed been investigated in a number of papers with a focus on computational efficiency and performance.
> > > As argued in the paper, while the performance was part of our objective, the efficiency in our case was a byproduct, which is clearly welcome.
> > > Our different perspective, clearly inspired from Physics and Information Theory, was presented as a potential new window on thinking about the diffusion problem. The theoretical underpinnings were provided to help drive the intuition moving forward.
> > > We have clearly answered the Reviewers’ concerns, answered questions and added   the requested comparative results to validate the soundness of the approach, and hope that the paper be viewed as a means of providing a different viewing angle of the problem with at least no loss in some cases and gains on performance in others. Please see other replies about how other computational engines can be used on the proposed methodology.

---

> ### Comment · Reviewer_T5eD · 2024-11-26
>
> Thank you once again for the rebuttal and for addressing my concerns.
>
> I have reviewed your response. Regarding Q2, Q3, and Q4, thank you for the clarification and for updating the manuscript.
>
> For Q2, I realize there was a misunderstanding on my part regarding the comment on the 'significant figures in Tables 1 and 2 being inconsistent.' Please disregard this point.
>
> However, I still think that the following point has not been adequately addressed. Regarding Q1, there is insufficient consideration of existing related research [1, 2]. These studies discuss not only adjustments in backward diffusion, such as with DPM solvers, but also approaches to handling noise scheduling in forward diffusion.
>
> [1] Chen, Ting. "On the importance of noise scheduling for diffusion models." arXiv preprint arXiv:2301.10972 (2023).
>
> [2] Chen, Yuzhu, et al. "Adaptive Time-Stepping Schedules for Diffusion Models." The 40th Conference on Uncertainty in Artificial Intelligence.

---

> ### Author Response · Authors · 2024-11-27
>
> Thank you for agreeing with our clarification. The noise schedules presented in [1] and [2] remain constant and uniform across all the pixels. The variance changes occur across time-steps. We note and would like to emphasize that our work has been focused on applying unequal diffusion rates across different pixels as well. i.e., the spatial (across data) non-uniform diffusion impacts the dynamics of the underlying diffusion and hence of the associated dynamical equations. This is a novel approach that has not been proposed, let alone explored, in earlier papers. **The noise schedules of [1] and [2], are distinct from our approach as they are discussing modulating diffusion rates across time, not pixels. We have included the references on account of their interesting discussion on scheduling across time-steps.**

---

### Official Review · Reviewer_enFG · 2024-10-26

**Soundness:** 2
**Presentation:** 1
**Contribution:** 2
**Rating:** 3
**Confidence:** 4

**Summary:**

This paper introduces a new diffusion model that diffuses each pixel at a different rate based on its value, inspired by the water pouring algorithm. With a different forward and backward process than traditional DDPM or SDE-based methods, the authors show that their method can reduce the number of sampling steps at inference time. The authors evaluate their method on the CelebA and CIFAR datasets.

**Strengths:**

- The idea of considering the value of each pixel to determine the drift and diffusion rate, inspired by traditional water pouring algorithm, seems interesting and worth studying.

**Weaknesses:**

- The motivation behind pixel-aware diffusion seems not clear to me. The water pouring algorithm is originally designed for communication systems but I am not sure how it benefits visual media such as image generation.
- This paper does not show how the new forward and backward processes look like, which would be useful for readers to understand better the motivation and how the proposed method changes visually the diffusion process.
- The writing and presentation of the paper can be greatly improved. For example, Figure 1 can be horizontal to have more spaces and can be easier to understand if the time-axis is put horizontally. Figure 2 is distorted. Figure 4 and the surrounding text are both not clear to me how the architecture is changed. Figure 5 should be shown without distortion and be following the style of Figure 7 and 8. Table 1 and 2 can be placed horizontally to save some space. In Line 194-195, DDIM should be from Song et al. In line 340, "on the hand" should be "on the other hand"? In 404-405, "Song et al." is repeated and the DDIM citation needs to be corrected. In appendix D line 5, it seems to introduce unnecessary symbols like the L for reverse diffusion noise predictions, which is not consistent to other works. Some citations seem missing, e.g., I did not find CelebA in the reference.
- To my understanding, the paper "Common Diffusion Noise Schedules and Sample Steps are Flawed" is related to this work. Their paper also introduces the idea of zero-snr and it makes sense to discuss the difference and even compare with it.
- The main results shown in table 1 and 2 seem not fair to me, especially on the number of sampling steps.

**Questions:**

- In table 1 and 2, why all settings (e.g., number of parameters and steps) are different? I can see the network architecture is modified in Section 3.3, but it is not clear why it is around 2x larger.
- The proposed method can reduce the number of steps from 1000 to 200. I think DDIM can also reduce to 200 with minimal loss of quality. However, the comparisons do not show how DDIM works with the same number of sampling steps, is there any reason for this?
- Why it is worth mentioning the continuous-time version in eq. 10, is it used later?
- In Figure 4, which part of the architecture is different from VAE and U-Net that are widely used in other diffusion models?
- In appendix B, the proof of convergence seems interesting. It discussed that we can carefully choose $\gamma$ to ensure effective pixel-wise diffusion, but is there ablation study to support that?

---

> ### Author Response · Authors · 2024-11-20
> **Response to reviewer's quesries**
>
> We thank Reviewer for their time and very valuable comments, as well as thankfully agree that the proposed approach is interesting and worthy of further study. We have carefully addressed all questions and concerns below. We will happily provide further clarification if needed.
>
> **Q1** I am not sure how Water Pouring Algorithm (WPA) benefits visual media such as image generation.
>
> Reply: The aim of any forward diffusion process is a destruction of information in an image by progressive “ablation” through the SNR reduction of each pixel to 0 (i.e., incremental addition of noise). Pixels with higher starting value should require more noise at each iterative step to globally reach 0 SNR together with lower value pixels. Unequally valued pixels undergo unequal noise addition. This is akin to WPA, except with an inverted objective. In WPA, we add higher amplification (i.e. signal power) to a channel with more noise to maximize total SNR. In our case, we add more noise to a channel (i.e. pixel) with higher value to minimize the SNR.
> Conventional diffusion models simply add the same noise power to each pixel. This lengthens the process with increasingly long diffusion steps for reaching 0 SNR at convergence. Our design achieves the convergence state at a faster rate than conventional models.
>
> **Q2** How the new forward and backward processes look like?
>
> Reply: We will add images of diffusion steps to show the progression our forward and reverse processes.
>
> **Q3** Typo corrections.
>
> Reply: Points well taken. We have made corrections and will upload an updated submission shortly. Thank you for the suggestions.
>
> **Q4** In appendix D line 5, it seems to introduce unnecessary symbols like the L for reverse diffusion noise predictions.
>
> Reply: L is the stylized Z in the LHS of Eqn.-14. We have changed the notation of noise predictions to remove any ambiguity.
>
> **Q5**	In table 1 and 2, why all settings (e.g., number of parameters and steps) are different? I can see the network architecture is modified in Section 3.3, but it is not clear why it is around 2x larger.
>
> Reply: Our network is larger on account of
> 	1) Our inclusion of the parameters of the scale estimator VAE, when counting the parameters.
> 	2) Simultaneous generation of noise prediction outputs (Eq. 14) for all time steps in comparison to the conventional models’ single-step output noise prediction.
>
> **Q6** The comparisons do not show how DDIM works with the same number of sampling steps, is there any reason for this?
>
> Result: Using the DDIM results mentioned in the original DDIM paper, it is understood that the generation quality is proportional to the number of steps in the diffusion process. The same is true with the reduced FID values in DDIM case, per shown results with 10, 100 and 1000 steps. The FID for a 500 step-model would be somewhere between those for 100 and 1000 step-model. Tables-1 and 2 show that the FID of our model is lower than the FID for 1000 steps conventional DDIM based model. Hence, our FID would also be lower than a 500 steps DDIM based conventional model. We have noted this in the third paragraph of our results section in our original submission.
>
> **Q7** Why it is worth mentioning the continuous-time version in eq. 10, is it used later?
>
> Reply: Please refer to our reply to Reviewer TBgv (Q2)
>
> **Q8** Which part of the architecture is different from VAE and U-Net that are widely used in other diffusion models?
>
> Reply: The fusion of information hidden in $x_{\delta}$ and the parallel prediction of noise outputs for all the steps (per Eqn-14) underlines the differences with a conventional UNET based design. So, for an RGB image, while conventional U-Net models have a 3-channel output, our model has a 3 x 500 channels output for a 500-time steps trajectory. Additionally, just like fusion of time step embedding in conventional models, image structure information in the form of $x_{\delta}$ is also fused inside the U-Net.
>
> **Q9** In appendix B, the proof of convergence seems interesting. It discussed that we can carefully choose $\gamma$ to ensure effective pixel-wise diffusion, but is there ablation study to support that?
>
> Reply: In the case of a single pixel, we have an exponential function that we can illustrate using the pixel value. We have also compared the conventional diffusion models with our design by conducting experiments with toy examples, where noise added in the forward direction is completely known beforehand. Specifically, we conducted reverse trajectory generation experiments where the entire forward direction trajectory (i.e., the noise samples Z in Step-5 of the Sampling algorithm-1 (Page-15) ) was known beforehand.  In Sampling algorithm-1 the scale estimate $\hat{x}$$\delta$ was replaced with actual scale value $x_{\delta}$ and we checked the MSE between generated reverse direction image, and the real image $x_0$. Appropriate value of $\gamma$ was chosen based on these experiments.

---

> > ### Comment · Reviewer_enFG · 2024-11-23
> >
> > I would like to thank the authors for making clarifications for Q1, Q3, Q5, Q6, Q7, Q9, as well as updating the paper.
> >
> > If I look at the FID numbers on CelebA, CIFAR10, they are actually good. However, I find it difficult to follow the details of the method, like the ones for Q4, Q8:
> > - The forward process is clear to me, but the reverse process (section 3.3) is not. To my understanding, section 3.3 should refer to Algorithm 1 and explain in details.
> > - Then, it's not well-motivated why model should output all the 500 noise predictions at once, and why the approximation of $x_\delta$ with only structural information is eough.
> > - Adding images of diffusion steps to show the progression of forward and reverse processes would help as discussed in Q2.

---

> > > ### Author Response · Authors · 2024-11-24
> > >
> > > Thank you for your reply and further appreciation of our work.
> > >
> > > Regarding **The forward process is clear to me, but the reverse process (section 3.3) is not. To my understanding, section 3.3 should refer to Algorithm 1 and explain in details.**
> > >
> > > **Reply**: We have simplified the notations of various blocks in the reverse diffusion model in Section-3.3. This is also reflected in the architecture figure, with more details provided in Appendix C. We have updated these changes in our latest available submission. Please go through Section-3.3 and Appendix C for exact details. As stated in the last paragraph of Section-3.3, at the time of inference, our sampling algorithm is quite similar to the one used in the original DDPM paper (Algorithm 2 in Ho et al 2020), with the differences already elaborated in Section 3.3 and the Algorithm-1 itself.
> > >
> > >
> > > Regarding **Adding images of diffusion steps to show the progression of forward and reverse processes would help as discussed in Q2.**
> > >
> > > **Reply:** We have added diffusion progression images in Appendix-G. They also give visual evidence of how our diffusion process results in faster convergence in comparison to conventional diffusion process.
> > >
> > >
> > > Regarding **Then, it's not well-motivated why model should output all the 500 noise predictions at once, and why the approximation of x_\delta with only structural information is eough.**
> > >
> > > **Reply:**Our approach does not use any DPM Solver or any other subsampling based accelerator, as our focus here is on the novel diffusion process. This necessitates us to generate all 500 time-steps. As we note in the first paragraph of Section-3.3, theoretically based on Universal Approximation Theorem, we can also add more parameters to a conventional reverse diffusion neural network  to predict all the steps in parallel, except for the fact of  making the network prohibitively large. In our case, at each time-step, we have partial information about the clean image $x_0$ available to us in the form of $x_{\delta}$. In contrast to our diffusion process, no such information is available in the case of conventional diffusion processes **at each time-step**. Availability of this partial information prior is the rationale for predicting all the 500 steps in parallel using a parsimoniously designed model.
> > >
> > > This partial information **acts as a regularizer** to our model and provides us with the opportunity to design the noise predictor $\mathscr{G}_j$ (the green colored box in the architecture image) with comparatively less number of parameters, effectively. This facilitates in accelerating the sampling speed through just a single forward pass through the neural network.
> > >
> > > Please also refer to Appendix F which further justifies our decision of using $x_{\delta}$ as an appropriate prior to regularize our model.

---

> > > > ### Comment · Reviewer_enFG · 2024-11-25
> > > >
> > > > Thanks for providing the diffusion process in Appendix G. I am still not convinced about the correlation between forward and backward. The forward corrupts the image rapidly, while the reverse is recovering only the image structure at most steps. It's difficult to say if it is converging faster. I would like to keep my score.

---

> > > > > ### Author Response · Authors · 2024-11-27
> > > > >
> > > > > In reverse, we are not recovering image structure using our parallelized reverse diffusion. Rather, we are recovering it in the form of $\hat{x}$$\delta$  using our scale estimator VAE. This recovered structure is **then** used as a prior in our reverse diffusion mode (given as input to the red colored blocks as shown in the architecture Fig.-4) to generate the noise predictions for all time-steps in parallel. The faster convergence in reverse is in the form of reduced total number of time-steps due to faster convergence in forward direction itself. For instance, in a conventional model, if 1000 steps are required in forward direction diffusion, then reverse diffusion models also use 1000 forward passes (assuming no accelerator or subsampler like DPM Solver or DDIM is used). In our case, we have shown (using Proposition 2 and also verified in Fig.-2), that using our diffusion process yields a 1000 time-step reduction to 500. So, our parallelized reverse diffusion model also needs to output noise predictions for just 500 time-steps instead of 1000 time-steps. Finally, due to our parallelized model (facilitated by the availability of coarse structure of clean image in the form of $\hat{x}$$\delta$), we get further acceleration in image generation.

---

### Official Review · Reviewer_dKU1 · 2024-10-29

**Soundness:** 2
**Presentation:** 2
**Contribution:** 2
**Rating:** 3
**Confidence:** 4

**Summary:**

This paper proposes to add uneven noise shedule to pixels according to their SNR, with higher SNR pixel diffuses faster than the lower one.

**Strengths:**

1. This paper proposes uneven noise diffusion of pixels according to their SNR.
2. The forward process under this paradigm is well formulated.

**Weaknesses:**

1. This paradigm seems restricted to pixel space, since it requires the SNR of per pixel for differentiated diffusion. This may severely limits the practicality of this method, since most of existing diffusion models adopt latent space for efficiency.
2. No proof for the claim. This paper claims less steps but lacks necessary evidences. Visualization or experiemtal results are rare.
3. Experimental results are quite few. This paper only conduct experiments on two small-scall/resolution datasets. Besides, the table1 and table2 are inprofession, where DDIM is sampler method instead of model.
4. Poor writing. This paper seems like a semi-finished product. The writing lacks necessary polishing and is diffucult for understanding.

**Questions:**

1. Is it possible to extend this method to latent space? The SNR value in latent space may be different from the pixel space, since the latent is trained with KL loss to be Gaussian.
2. Will the optimization difficulty become harder as the resolution getting larger. The uneven pixel noise schedule may bring high uncertainty with higher resolution.

---

> ### Author Response · Authors · 2024-11-20
> **Response to reviewer's queries**
>
> Thank you for taking time to review our paper and provide valuable comments. We have carefully addressed your questions and concerns in our following response. We will be glad to provide further clarification if you have any further concerns.
>
> **Q1** Is it possible to extend this method to latent space? The SNR value in latent space may be different from the pixel space, since the latent is trained with KL loss to be Gaussian.
>
> Reply: We agree there are multiple design additions similar to those employed with conventional models, which can be applied to our model to further improve the performance. However, our focus in this submission was more on presenting a novel diffusion process than presenting a full-fledged diffusion process competitor.
>
> As Eqns.-8 and 9, as well as Steps-2 and 3 of Algorithm-1 show, we do require the value of $\hat{x}$$\delta$ for calculating/generating $\alpha_i$’s, and eventually the complete reverse time trajectory. Using a latent space representation of $\hat{x}$$\delta$ (instead of pixel-space value) would require us to change our sampling algorithm. We would also have to add some parameterized learnable transformation of the latent space representation of $\hat{x}$$\delta$ to our present model and modify the training objective accordingly. We welcome this idea of reviewer. But to keep the trajectory generation step easier to analyze, in our initial design, we have constrained all the training over pixel-space.
>
> **Q2** Will the optimization difficulty become harder as the resolution getting larger. The uneven pixel noise schedule may bring high uncertainty with higher resolution.
>
> Reply: Indeed, with increased resolution, the model size would increase for all algorithms. However, in our case, with the help of our pixel-aware forward diffusion, the parameter increase factor is not significant. As is evident from the parameters count in Tables 1 and 2, moving from 32 x 32 to 128 x 128 resolution, only resulted in approximately 2x increase in the number of trainable parameters, which is not a significant an increase in comparison to conventional diffusion model. We expect a similar manageable increase in the case of larger resolutions.

---

> > ### Comment · Reviewer_dKU1 · 2024-11-25
> >
> > Thanks for the response. Indeed, the concerns of the applicability on latnet space and larger resolutions are not addressed. From resolution 32x32 to 128x128 is not representative, since 128 resolution is still far from usage. Besides, the concern of limited experiments is not replied. For now, I tend to maintain my score.

---

> > > ### Author Response · Authors · 2024-11-27
> > >
> > > On why we have relied on pixel space diffusion instead of latent space diffusion,
> > >
> > > 1) Our reverse diffusion model leverages the information provided by $\hat{x}$$\delta$  for its parallelization to be successful.  $\hat{x}$$\delta$ itself clearly carries coarse scale information about the image structure, providing each channel of the parallel output this **explicitly** available information about  $x_0$.
> > >
> > > 2) Our sampling algorithm also requires knowledge of  $\hat{x}$$\delta$ (please refer to Steps-2 and 3 in Algorithm 1).
> > >
> > > If we use an autoencoder on top of  $\hat{x}$$\delta$, to get a low dimensional embedding, say  $x_{\delta,e}$, then there are two possible disadvantages:
> > >
> > > i) Further increase in the number of parameters as a consequence of adding another autoencoder besides our scale estimator VAE.
> > >
> > > ii) We can't be sure that the latent space of   would also carry the same intended structural information as  , which we require in the green blocks of our architecture for generating noise predictions for all the time-steps.
> > >
> > > If we add an autoencoder for the noisy input $x_i$  only instead (before inputting it to our reverse diffusion model), then again we add extra parameters in our model.
> > >
> > > That being said, reviewer dKU1 actually raised a very good point that by combining with Latent method, our new diffusion can actually run on a higher resolution image, by swapping the conventional diffusion in the Latent diffusion model with our diffusion.  Soliciting Reviewer dKU1’s understanding our inability to run the experiment due to time constraint and provide in a timely fashion the results. We still hope the reviewer will agree  that  our new diffusion provides research colleagues a path to yet other new and  further developments.

---

### Official Review · Reviewer_TBgv · 2024-11-02

**Soundness:** 3
**Presentation:** 1
**Contribution:** 3
**Rating:** 3
**Confidence:** 4

**Summary:**

The authors introduce a new generative method inspired by the water-pouring algorithm paradigm as an alternative to existing diffusion methods, offering faster generative sampling (with fewer denoising steps, achieving a speedup of 4x) to produce high-quality images. Each pixel has a unique scheduling function b_i, based on the signal-to-noise ratio (SNR) of the pixel.

**Strengths:**

It is an interesting alternative to existing methods. Incorporating pixel values into the scheduler is not a straightforward solution, and using the SNR with the water-pouring algorithm seems novel. The model has been evaluated on CelebA and CIFAR-10, demonstrating lower FID scores and faster sampling speeds.

**Weaknesses:**

Current literature includes other models that achieve high performance with lower sampling speeds. DDIM was the SOTA algorithm 3-4 years ago, and many improvements have been made since then. I would suggest that the authors also compare with the following:

**Consistency Models**:
- Consistency Models ([https://arxiv.org/abs/2303.01469](https://arxiv.org/abs/2303.01469))
- Improved Techniques for Training Consistency Models ([https://arxiv.org/abs/2310.14189](https://arxiv.org/abs/2310.14189))

**DPM-Solvers**:
- DPM-Solver: A Fast ODE Solver for Diffusion Probabilistic Model Sampling in Around 10 Steps ([https://arxiv.org/abs/2206.00927](https://arxiv.org/abs/2206.00927))
Several papers have been published following this line.

Additionally:

- It is unclear why the authors use the discrete formalism of DDPM in the background section, then apply their solution in a continuous scenario (Appendix B).

- Where does Equation (8) come from? Is it based on an assumption?

- A challenging part of the method is obtaining \(\overline{\alpha} = \prod \alpha_t\). How is this achieved in Equation (9)? I consider it tricky because applying the chain rule to go from \(x_0\) to \(x_t\) is complex:

x_{t-1} = \sqrt{\alpha_t} \cdot x_t + \sqrt{1 - \alpha_t} \cdot \epsilon = \sqrt{\alpha_{t+1}} \cdot (x_{t+1} + \sqrt{1 - \alpha_{t+1}} \cdot \epsilon) = ...

If \(\alpha) depends on the signal, it becomes particularly hard to solve. Could you explain more clearly how these results were derived?

- **Figures**: ALL the figures need improvement. The text appears blurred and stretched, with inconsistent font sizes, making them hard to read. Figure 1, which presents the main concept of the paper, could be more impactful and should be revised to better communicate the work to the community. Figure 2 has small text on the axes and a stretched legend. Figure 3 does not add much context; the image in Figure 6 in the appendix is more interesting and from my point of view can substitute Fig. 3. Figure 4 is unclear due to color choices, labels, and structure. Figure 5a is difficult to interpret since the images are too small; it would be better to display a few images and compare them across different methods with the same seed.

The results are interesting, but I would only raise my score only if the presentation would be improved.

**Questions:**

Already in the weaknesses.

---

> ### Author Response · Authors · 2024-11-20
>
> We thank the reviewer for their time and valuable comments. We have carefully addressed all questions and concerns in what follows. We will happily provide further clarification for any further concerns.
>
> **Q1** Current literature includes other models that achieve high performance with lower sampling speeds. DDIM was the SOTA algorithm 3-4 years ago, and many improvements have been made since then.
>
> Reply: Point well taken. Please note that our work firstly at fundamentally reducing the forward diffusion path. While recent improvements are mostly focusing on speeding up the reverse diffusion, improvements on forward diffusion are limited. The novelty of our algorithm is in the proposed variable strength of the pixelwise diffusion, which effectively achieves faster diffusion. For a better perspective, we feel that it is good to update our comparison to include DPM solver, as stated in our reply to reviewer aRbR--please see our detailed comments on the SOTA algorithms, the table only shows the backward diffusion time.  Considering that our forward diffusion trajectory has been shortened, reverse trajectory is also consequently shortened, and its parallelization by our introduction of the image scale $x_{\delta}$ provides significant run-time reduction, while maintaining reasonable low FID.
> Comparison with DPM Solver applied to conventional DDPM for CelebA dataset:
> | Model                  | #Param.(M)       | #Diff Steps | FID   | Execution Time (sec) |
> |------------------------|------------------|-------------|-------|----------------------|
> | DDPM                   | 78.7             | 1000        | 3.51  | 10.19                |
> | SDE Based (Song et al) | 65.6             | 1000        | 3.20  | 246.69               |
> | DDIM                   | 78.7             | 10          | 17.33 | 0.53                 |
> | DDIM                   | 78.7             | 100         | 6.53  | 5.55                 |
> | DPM Solver             | 78.7             | 1000        | 3.51  | 48.44                |
> | DPM Solver             | 78.7             | 10          | 4.85  | 0.04                 |
> | DPM Solver             | 78.7             | 34          | 4.53  | 0.12                 |
> | DPM Solver             | 78.7             | 100         | 4.52  | 0.33                 |
> | DPM Solver             | 78.7             | 500         | 3.79  | 1.48                 |
> | Our Model              | 61.3+84.2 =145.5 | 500         | 3.25  | 1.3                  |
> Here our model includes parameters from the VAE as well as the parallel diffusion model.
>
> **Q2** It is unclear why the authors use the discrete formalism of DDPM in the background section, then apply their solution in a continuous scenario (Appendix B).
>
> Reply: Keeping with the conventional development of evolution of diffusion models, we have first developed a discrete time model. This paper only discusses the discrete time scenario. The continuous time version is a work in progress. Our invoking of the continuous case, serves to explain our algorithm and helps in clarifying its advantages. Specifically, we use in Appendix B, a continuous time scenario to exploit the well-developed SDE theory to show time evolution of the mean of forward trajectory (Please see Sarkka and Solin (2019) in the references). As a result, we have shown that we can achieve faster convergence in the forward direction in comparison to conventional models. This resulted in reducing the trajectory length in reverse direction as well, which helps in reducing the parameters of our parallelized reverse diffusion model. We will add a bridging sentence to highlight the rationale.
>
> **Q3** Where does Equation (8) come from? Is it based on an assumption?
>
> Reply: Eqn-8 is a definition of parameter $\alpha_i$. It is used to design our forward schedule. As stated in our original submission, line 215, we are able to achieve pixel-aware accelerated diffusion based on this forward schedule. We will further clarify in our revision.
>
> **Q4** A challenging part of the method is obtaining (\overline{\alpha} = \prod \alpha_t). How is this achieved in Equation (9)? I consider it tricky.
>
> Reply: Point well taken. The derivation of Eqn-9 is similar to the one followed in conventional designs. Please refer to the derivations in the ‘Forward diffusion process’ section in this link: https://lilianweng.github.io/posts/2021-07-11-diffusion-models/

---

> ### Comment · Reviewer_TBgv · 2024-11-22
>
> Q1 I would like to thank the authors for their effort regarding expanding the benchmarks with the frameworks that I pointed out.
>
> Q4 I don't think that the derivation is similar to the basic one since the noise becomes dependent on the signal while the base derivation described in lilianweng's blogpost refers to the DDPM paper and in that case the noise is independent. I would like to have some proofs of that
>
> Perhaps, the main problem of this paper is the whole presentation. Even thought the idea is interesting, the presentation has several issues. I still will keep my score 3 (rejected) because the figures are not for an high-quality standard publication. I listed in the previous answer all my concerns about it.

---

> > ### Author Response · Authors · 2024-11-23
> >
> > Thank you for your reply and accepting most of our explanation!
> >
> > In regard to your Q4, while our proposed diffusion is pixel-wise aware,  the derivation of Lilian's blog still works.  The diffusion at each pixel is occurring independently of each other, due to element-wise multiplications in Equation-9. Each diffusion has its noise that is independent from other pixels. We hope this answers the reviewer's question.
> >
> > We have also made the fonts and legends of all the images clearer (including architecture details for better understanding). We have incorporated these changes in our edited submission, now available for everyone's review.
> >
> > Please let us know if you have any more questions.

---

### Official Review · Reviewer_aRbR · 2024-11-04

**Soundness:** 2
**Presentation:** 3
**Contribution:** 2
**Rating:** 3
**Confidence:** 4

**Summary:**

This paper proposes a modification to the forward process of a diffusion model by varying the diffusion schedule for each pixel based on that pixel's initial value. This results in faster convergence to Gaussian white noise than the standard forward diffusion process, which is not pixel-aware. Then, in order to perform the reverse process during inference, an auxiliary model is used to predict the per-pixel noise schedule at each time step. This formulation allows sampling from the diffusion model with fewer time steps and additionally makes it possible to parallelize across time steps, generating the entire reverse trajectory in a single forward pass of the model.

**Strengths:**

The overall observation that the diffusion schedule can be pixel-dependent is a reasonable one, and some of the theoretical analysis showing the connection between $x_0$ and the corresponding PSNR trajectory is interesting. Additionally, the fact that this enables parallelized one-shot inference is a nice extension of the approach. Overall, the goal of accelerating diffusion is valuable, and this paper makes some progress to that end.

**Weaknesses:**

My main concerns have to do with limited comparisons to past work, sensitivity to hyperparameters, and insufficient justification/ablation. Specifically, the method is not compared to any SOTA approaches to diffusion acceleration, hyperparameter choices are not justified, and the sensitivity of the algorithm to the approximation of $x_\delta$ is not sufficiently analyzed. For more details see below --- it would be great for the author's to provide a response to these points.

Minor typos/issues:

L151: "offering" -> "offer

L160: "Gallager" \citep

L183: "SNRs 6 time steps" -> "SNRs over 6 time steps"

L191: "Furthermore, the" -> "Furthermore,"

L196 "over-all" -> "overall", "we proposed" -> "we propose"

L326: "atleast" -> "at least"

Figure 4 is rasterized and aliased

**Questions:**

How does this approach compare, both in terms of results and theoretically, to the many other works on diffusion acceleration? This includes distillation-based approaches such as "Consistency models" [Song et al. ICML 2023], sampler-based approaches (e.g.,"DPM-Solver" [Lu et al. NeurIPS 2022]), as well distribution-matching approaches ("One-step Diffusion with Distribution Matching Distillation" [Yin et al. CVPR 2024]). None of these or other related works are cited or discussed despite being very closely related. The only comparisons in the paper are to DDIM, DDPM, and SDE-based settling, which do not accurately reflect the SOTA.

Additionally, this approach is only demonstrated for pixel-space diffusion models, while many of the other works in diffusion model acceleration operate on latent space models, which are a more practical setting. Does the proposed approach generalize to latent diffusion models?

I am also wondering how sensitive the proposed algorithm is to hyperparemeters and how hyperameters were chosen in the paper. In particular, Section 3.2.1 mentions considerably different $T$ and $\gamma$ values chosen for the two datasets that the model was evaluated on. Is there a systematic approach to picking these values?

It would be good to include some additional analysis on how well the VAE is able to approximate $x_\delta$ (as opposed to $x_0$) and to what extent this error affects the result of the inversion diffusion. Figure 3 attempts to visualize this, but is difficult to parse and is not especially informative. In particular, the author claim that $x_\delta$ "lacks finer details" compared to $x_0$ due to the large $\gamma$, making it easier to approximate. However, isn't this a function of the timestep, i.e., for earlier timesteps, the value of $\gamma$ must be lower? I would be curious to see some elaboration on this topic, since it is a critical piece of the proposed method.

---

> ### Author Response · Authors · 2024-11-20
> **Response to reviewer's queries**
>
> We thank the reviewer for their time and very valuable comments. We have carefully addressed all the questions and concerns below.  We will happily provide further clarification if needed.
>
> **Q1** On Typos and figure clarity:
>
> Reply: Point well taken. We have fixed them in the revision and the updated file will be up shortly.
>
> **Q2** Comparison with SOTA models.
>
> Reply: Our updated table here shows a comparison with DPM Solver applied to conventional DDPM for CelebA dataset. Considering that our forward diffusion trajectory has been shortened, reverse trajectory is also consequently shortened, and its parallelization by our introduction of the image scale $x_{\delta}$ provides significant run-time reduction, while maintaining reasonable low FID. This further supports the reviewer comment, which we agree with,: “the goal of accelerating diffusion is valuable, and this paper makes some progress to that end”. Current models have only focused on reducing the reverse trajectory by employing sub-sampling or fast ODE solver based strategies (also applicable to our model) without trying to reduce the forward trajectory.
>
> CelebA performance
> | Model                  | #Param.(M)       | #Diff Steps | FID   | Execution Time (sec) |
> |------------------------|------------------|-------------|-------|----------------------|
> | DDPM                   | 78.7             | 1000        | 3.51  | 10.19                |
> | SDE Based (Song et al) | 65.6             | 1000        | 3.20  | 246.69               |
> | DDIM                   | 78.7             | 10          | 17.33 | 0.53                 |
> | DDIM                   | 78.7             | 100         | 6.53  | 5.55                 |
> | DPM Solver             | 78.7             | 1000        | 3.51  | 48.44                |
> | DPM Solver             | 78.7             | 10          | 4.85  | 0.04                 |
> | DPM Solver             | 78.7             | 34          | 4.53  | 0.12                 |
> | DPM Solver             | 78.7             | 100         | 4.52  | 0.33                 |
> | DPM Solver             | 78.7             | 500         | 3.79  | 1.48                 |
> | Our Model              | 61.3+84.2 =145.5 | 500         | 3.25  | 1.3                  |
> Here our model includes parameters from the VAE as well as the parallel diffusion model.
>
> **Q3** Does the proposed approach generalize to latent diffusion models?
>
> Reply: Point well taken. The latent space model, with some special care given to the image scale, should be able to swap the conventional diffusion with our new model.  We will include this comment in the updated resubmission to clarify our contribution. As noted earlier, our focus is more on a novel diffusion model.
>
> **Q4** Is there a systematic approach to picking the values for T and $\gamma$ ?
>
> Reply: Our pixel aware forward diffusion keeps T smaller than that in conventional DM. Larger $\gamma$ results in faster convergence to standard normal distribution in the forward direction and provides image scale without detailed features for reconstruction. Too large a value could break the Markovian characteristic of the forward trajectory, making the task of denoising in reverse direction unnecessarily difficult. Note that, we opted to keep a 1:10 ratio between $\gamma$ and T. While heuristically chosen, they followed a thorough experimental validation over toy examples. We have some discussion on the choice of the parameters in Section 5/pg. 8 of the original submission. Additionally, due to lower information content in lower resolution images, as in CIFAR10 dataset, the forward diffusion tunes in to noise at a much faster rate in comparison to when using higher resolution CelebA images. Additional clarification on the hyperparameter selection will be given in the resubmission.
>
> **Q5** I would be curious to see some elaboration on how well the VAE is able to approximate $x_{\delta}$ (as opposed to $x_0$)
>
> Reply: The image scale, $x_{\delta}$ and $\gamma$ are independent of time-step (while $\overline{\alpha}$ ’s are dependent). Our scale approximator VAE learns to sample from the distribution of  $x_{\delta}$ to estimate it. We will include an example contrasting a clean image, $x_0$ and $x_{\delta}$ to provide a clearer comparison between the two.
>
> We have also trained an autoencoder sampling from the distribution of $x_0$ with the same number of parameters and architecture as our scale approximator VAE, to show that it is easier to learn to sample from the distribution of $x_{\delta}$ than from the distribution of $x_0$. As will be evident from example outputs from this VAE, its outputs have various unnecessary artefacts. We will update our resubmission with this information.
>
> As $x_{\delta}$ works as an informational prior in our reverse generative model, without unnecessary spurious information thus reducing their complexity. $x_{\delta}$ with only coarse level information about the image structure, is a better-informed prior.

---

### Author Response · Authors · 2024-11-23
**Updated paper uploaded**

We thank all the reviewers for their constructive comments. Based on these comments, we have made revisions to our submission (including changes to figures and results tables) and uploaded the updated paper. The new edits are in blue inked text.

Thank you again!

---

### Author Response · Authors · 2024-11-24
**Reverse Diffusion Model description simplified**

Hello all,

To make the description of our parallelized reverse diffusion model clearer, we have simplified the notations of various components of our architecture. The same is updated in Section 3.3 and in Appendix-C, **with more description added**. We request you to take a look at these sections for exact details.

We have also added examples of diffusion process progression in Appendix-G to further show how our diffusion process converges faster in comparison to conventional diffusion process.

We have uploaded the updated paper now with these changes.

Thank you again!

---

### Author Response · Authors · 2024-12-03

We thank all the reviewers again for their comments and suggestions. As the review period is coming to end, we invite all the reviewers to take a look at our final edited paper as well as our comprehensive answers to their questions.

---

### Meta-Review · Area_Chair_fgHd · 2024-12-16

**Metareview:**

This paper introduces a novel diffusion model that employs a pixel-value-dependent diffusion rate, drawing inspiration from the water-pouring algorithm. By utilizing distinct forward and backward processes compared to traditional DDPM or SDE-based methods, the authors aim to reduce the number of sampling steps required during inference.

However, the paper suffers from several shortcomings:

- Unclear Motivation: The rationale behind pixel-aware diffusion is not well-articulated. The authors fail to convincingly explain why this approach is beneficial and how it addresses limitations in existing methods.
- Incomplete Presentation: The paper reads like a work in progress. Crucially, it lacks a clear description of the proposed forward and backward processes. Visualizing these processes would significantly aid in understanding the method's impact on the diffusion process.
- Insufficient Experimental Validation: The experimental evaluation, limited to CelebA and CIFAR datasets, does not provide compelling evidence to support the claims of accelerated sampling. More extensive experiments on diverse datasets are needed.
- Lack of Clarity: The writing quality is poor, hindering comprehension. The paper requires substantial revision to improve clarity and presentation.

While the proposed method holds potential for accelerating diffusion models, the paper in its current form falls short of convincingly demonstrating its effectiveness. A more thorough and rigorous treatment is necessary to fully explore the potential of this approach.

**Additional Comments On Reviewer Discussion:**

The reviewers raised several concerns during the discussion period, including:

- Lack of clarity and motivation for the proposed method.
- Limited comparison to past work.
- Sensitivity to hyperparameters; Insufficient justification/ablation.
- Poor presentation and writing quality.

The authors addressed some of these concerns by providing additional explanation and making some changes to the paper. However, some concerns remain, such as the lack of clarity and motivation for the proposed method, and the limited comparison to past work.

In my final decision, I weighed the strengths and weaknesses of the paper, as well as the rebuttal discussion. I also considered the potential impact of the proposed method. Overall, I believe that the paper has some interesting ideas, but it is not yet ready for publication. The authors need to address the remaining concerns and improve the presentation and writing quality before the paper can be accepted.

---

### Decision · Program_Chairs · 2025-01-22

Reject